# ASoP (v1.0): A set of methods for analyzing scales of precipitation in general circulation models

Nicholas P. Klingaman[1], Gill M. Martin[2], and Aurel Moise[3]

[1]National Centre for Atmospheric Science–Climate and Department of Meteorology, University of Reading, Earley Gate,
P.O. Box 243, Reading, Berkshire RG6 6BB, United Kingdom
[2]Met Office, Exeter, United Kingdom
[3]Bureau of Meteorology, Melbourne, Australia

*Correspondence to:* Nicholas P. Klingaman (nicholas.klingaman@ncas.ac.uk)

**Abstract.** General circulation models (GCMs) have been criticized for their failure to represent the observed scales of precipitation, particularly in the tropics where simulated daily rainfall is too light, too frequent, and too persistent. Previous assessments have focused on temporally or spatially averaged precipitation, such as daily means or regional averages. These evaluations offer little actionable information for model developers, because the interactions between the resolved dynamics
and parameterized physics that produce precipitation occur at the native gridscale and timestep.

We introduce a set of diagnostics (ASoP1) to compare the spatial and temporal scales of precipitation across GCMs and observations, which can be applied to data ranging from the gridscale and timestep to regional and sub-monthly averages. ASoP1 measures the spectrum of precipitation intensity, temporal variability as a function of intensity, and spatial and temporal coherence. When applied to timestep, gridscale tropical precipitation from ten GCMs, the diagnostics reveal that far from the
10 "dreary" persistent light rainfall implied by daily mean data, most models produce a broad range of timestep intensities that span 1–100 mm day$^{-1}$. Models show widely varying spatial and temporal scales of timestep precipitation. Several GCMs show concerning quasi-random behavior that may influence alter the spectrum of atmospheric waves. Averaging precipitation to a common spatial ($\approx$600 km) or temporal (3-hr) resolution substantially reduces variability among models, demonstrating that averaging hides a wealth of information about intrinsic model behavior. When compared against satellite-derived analyses at
15 these scales, all models produce features that are too large and too persistent.

## 1 Introduction

Advances in supercomputing power continue to enable refinements in the resolutions of general circulation models (GCMs) used to simulate the effects of climate variability and anthropogenic climate change. As GCMs have become better able to resolve regional-scale boundary features (e.g., orography, coastlines), the scientific community has paid increasing attention
to these models' representations of local and regional hydrological extremes (e.g., Dai, 2006; Wilcox and Donner, 2007; Rosa and Collins, 2013), including the sensitivity of those extremes to climate change (e.g., Trenberth, 2011; Kharin et al., 2013; Pendergrass and Hartmann, 2014; Westra et al., 2014). Robust projections of local and regional changes in extremes with anthropogenic warming are essential to underpin decisions on adaptation strategies; accurate predictions of these extremes in

response to natural climate variability are critical for preserving lives and livelihoods, for example through emergency response and anticipatory aid efforts, particularly on sub-seasonal–seasonal scales.

Despite refinements in resolution and efforts to revise the treatment of sub-gridscale processes such as deep convection, climate models are criticized routinely for their inability to represent the observed frequency, intensity and persistence of
precipitation. Dai (2006) compared daily precipitation in 18 GCMs from the Third Coupled Model Intercomparison Project (CMIP3) against satellite-derived analyses from the Tropical Rainfall Measuring Mission (TRMM) dataset across 50°S–50°N. The CMIP3 models produced precipitation too frequently, particularly light precipitation ($< 10$ mm day$^{-1}$), but did not produce heavy precipitation ($> 20$ mm day$^{-1}$) frequently enough. Models performed similarly poorly when compared against gridded gauge data over land (Sun et al., 2006). Wilcox and Donner (2007) obtained similar results at the sub-daily scale,
demonstrating that 30-min averaged rainfall (sampled every 3h) from the Geophysical Fluid Dynamics Laboratory model was biased towards low intensities relative to TRMM. Revisions to the convective parameterization, particularly the closure and the triggering function, increased heavy precipitation frequency and reduced light precipitation frequency. Stephens et al. (2010) employed observations from the CloudSat spaceborne cloud-profiling radar to show that although contemporary GCMs produced reasonable seasonal and annual precipitation accumulations, these accumulations arose from highly biased daily
precipitation distributions: models produced precipitation far too frequently and far too lightly. The strong preference for persistent, light daily accumulations led the authors to call the GCMs' simulated world "dreary". Such biases lead to erroneously large moisture recycling over land, with consequences for the simulation of the global hydrological cycle (e.g., Trenberth, 2011; Demory et al., 2014).

More recently, Koutroulis et al. (2015) found that GCMs from the Fifth Coupled Model Intercomparison Project (CMIP5)
had improved somewhat in their daily precipitation distributions relative to their CMIP3 counterparts, particularly through an increase in the frequency of intense precipitation and a reduction in the overall frequency of precipitation. Hirota and Takayabu (2013) showed improved skill for 1–5 day precipitation extremes in CMIP5 relative to CMIP3. However, Rosa and Collins (2013) concluded that CMIP5 GCMs still produced 3-hr rain rates of 1–10 mm day$^{-1}$ too frequently over the southeastern United States, compared to gridded gauge data. When the models did produce heavier events, those events were too persistent.
Although the studies above highlight a heightened focus on the GCM representations of hydrological extremes—which are inherently small-scale, short-lived features—most evaluation of GCM precipitation focuses on gross spatial (e.g., regional averages) and temporal (e.g., monthly and seasonal means) characteristics (e.g., Phillips and Gleckler, 2006; Bollasina and Ming, 2013; Li and Xie, 2014; Mehran et al., 2014). Where attention is paid to shorter-term variability, studies have adopted a phenomenological approach, analyzing precipitation associated with synoptic features such as mesoscale fronts and con-
vective systems (e.g., Brown et al., 2010; Catto et al., 2013; Van Weverberg et al., 2013) or sub-seasonal modes such as the Madden–Julian oscillation (MJO; e.g, Hung et al., 2013). Yet the processes that produce precipitation in GCMs—the interactions between the sub-gridscale parameterizations and the resolved dynamics—function on the native gridscale and timestep of the models, not on a 3-hr or daily mean basis or on a regional average. Although it is often hypothesized that biases in the distributions of spatially and/or temporally averaged precipitation are the result of errors at the gridpoint, timestep level, few
studies have examined the spatial and temporal characteristics of precipitation at these most fundamental scales. In isolated

single-column model experiments, convective parameterizations have been shown to produce highly intermittent timestep precipitation (e.g., Stirling and Stratton, 2012), but it is not clear how, or even if, this behavior influences the distributions of precipitation at larger and longer scales. Information about the spatial and temporal characteristics of gridscale precipitation are far more useful for informing parameterization development than information about regional biases in seasonal, or even daily, accumulations.

The dearth of studies focused on the gridscale and timestep may be due to a lack of data, since large GCM intercomparison efforts such as CMIP5 do not collect timestep output to limit the volume of data produced. However, a recent model-evaluation project focused on the MJO (Klingaman et al., 2015) collected timestep data from ten GCMs for a limited number of short hindcast simulations (Xavier et al., 2015). Xavier et al. (2015) found that models differed considerably in the degree of timestep-to-timestep precipitation variability over a $5° \times 5°$ region of the equatorial Indian Ocean, as computed by the root-mean-squared difference of area-averaged timestep precipitation, but did not connect this variability to other scales or examine the spectra of rainfall intensities. There was no relationship between timestep precipitation variability and MJO fidelity (Klingaman et al., 2015).

Another reason for the lack of attention to timestep precipitation may be a scarcity of suitable diagnostics to compare the characteristics of precipitation variability among models, and between models and observations, across spatial and temporal scales. Previous studies have focused mainly on frequency distributions of precipitation intensities, computed mainly at the model gridscale but often on time-averaged or selectively sampled data (e.g., one 30-min model timestep per 3-hr). While these results are useful, they do not consider the coherence of precipitation features in space and time. Diagnostics of precipitation coherence require sampling many gridpoints and timesteps, which can be computationally cumbersome when working with high-frequency, fine-resolution data.

In this manuscript, we introduce diagnostics designed to describe precipitation variability and scale interactions in observations and models across a range of spatial and temporal scales. These diagnostics were developed with a view to condensing large data volumes from sub-daily output of O(10)km-scale GCMs into a set of measures of precipitation frequency, intensity and spatial and temporal coherence, to improve understanding of observed rainfall variability and compare simulated and observed precipitation characteristics across a range of scales. The diagnostics form a small software package entitled "Analyzing Scales of Precipitation", version 1.0 (ASoP1). In section 2, we describe the ASoP1 diagnostics, then introduce the MJO hindcast dataset mentioned above. In section 3 we demonstrate how the diagnostics can be used to discern and evaluate model behavior by applying them to the MJO hindcast dataset as well as to satellite-derived precipitation analyses. We discuss our results in section 4 and summarize our findings in section 5.

## 2  Diagnostics and datasets

As this manuscript focuses on novel diagnostics of precipitation, we devote section 2.1 to a thorough explanation of our methods, including examples using satellite-derived precipitation analyses. To demonstrate the ability of these diagnostics to compare precipitation characteristics among a range of model configurations, we apply the ASoP1 diagnostics to sub-daily

tropical precipitation from ten models from the "Vertical structure and physical processes of the Madden–Julian oscillation" model-evaluation project, described in section 2.2 and shown in Table 1, as well as 3-hr data from two satellite-derived analyses: TRMM 3B42 product, version 7A (Kummerow et al., 1998; Huffman et al., 2007, 2010, ; hereafter "TRMM") and the National Oceanic and Atmospheric Administration Climate Prediction Center Morphing Technique, version 1.0 (Joyce et al., 2004, ; hereafter "CMORPH"). We employ two observation-based datasets to provide a measure of observational uncertainty in our diagnostics. Both products are derived from a combination of infrared and microwave sounders. The TRMM 3B42 algorithm combines available precipitation data from microwave sounders, then fills the gaps in this dataset by merging precipitation data from infrared sounders, which have been first calibrated against the microwave sounders over a longer time period. TRMM 3B42 is calibrated on a monthly basis against gauge data from the Global Precipitation Climatology Centre. The 3B42 algorithm uses many microwave sounders, not only the TRMM Microwave Imager; "TRMM" refers to the source of the data, not the instrument. CMORPH combines precipitation data from microwave sensors only, but fills the gaps between microwave satellite overpasses by advecting the precipitation field using vectors derived from infrared-based cloud observations. Both products have been shown to under-detect light rainfall rates (e.g., Huffman et al., 2007; Tian et al., 2010). We use TRMM and CMORPH in the domain 60°–160°E, 10°S–10°N for two periods in boreal winter 2009–10, the choice of which is described in section 2.2.

TRMM and CMORPH have a native horizontal resolution of 0.25°×0.25°, which is finer than any of the models analyzed. Because the diagnosed spatial and temporal scales of precipitation will vary with horizontal resolution, we use an area-weighted averaging method to interpolate TRMM and CMORPH to a 1.25°×1.25 grid, which is approximately the median resolution of the models (147 km). A robust validation of any one model would require averaging TRMM and CMORPH to the model's native resolution, or preferably to a common resolution coarser than the model's native grid, as our results suggest. However, model validation is not the purpose of our study, so for clarity of presentation we compare the models to the 1.25° TRMM and CMORPH data to indicate observed scales of precipitation at a resolution comparable to, but not exactly equal to, the models' resolution. The example diagnostics below demonstrate the effects of horizontal resolution on the scales of precipitation, using 0.25° and 1.25° CMORPH data.

We discuss the diagnostics first, as they are designed to be applied to any model or observed dataset at scales ranging from the model timestep to a sub-seasonal average, and from the gridscale to O(1000 km) regions, depending on the phenomena and scales of interest. The results we show in section 3 for timestep and 3-hr precipitation are only one example use of these diagnostics. In all our diagnostics, we scale precipitation rates to $mm\,day^{-1}$, since this units is commonly used in other studies. However, it should be remembered that a fixed value in $mm\,day^{-1}$ equates to various rainfall intensities depending on the temporal scale considered (e.g., a 20-min timestep or a 3-hr average).

## 2.1 Methods

### 2.1.1 Precipitation spectra and contributions to total precipitation

To examine the precipitation intensity distribution on a given temporal or spatial scale, and its sensitivity to temporal and spatial averaging, we compute the contributions of discrete bins of precipitation intensity to the total precipitation at a gridpoint. These contributions can be expressed as either a precipitation rate, where the sum across all bins gives the total precipitation rate, or as a fraction of the total precipitation rate, where the sum across all bins is unity. In the latter case, the result is a spectrum that shows the relative importance of precipitation events in a given intensity bin to the total precipitation, while the former also includes contributions from the frequency of each precipitation rate. We use 100 bins ($b$; mm day$^{-1}$), for which the edges are defined by:

$$b_i = e^{\left\{ \ln(0.005) + \left[ i \cdot \frac{(\ln(120) - \ln(0.005))^2}{59} \right]^{\frac{1}{2}} \right\}} \tag{1}$$

where $i$ is the number of the bin and $\ln(x)$ is the natural logarithm of $x$. We add a further lower bin edge at 0.0 to ensure that a histogram computed using these bins sums to the number of valid data points in the sample.

The calculations can be performed for any input grid and temporal resolution. By calculating these contributions at each gridpoint in a region, we produce maps of the contributions of precipitation intensity bins to the total precipitation at each gridpoint. Examples of these for 3-hr TRMM and CMORPH 1.25° data are shown in Fig. 1. These contributions can then be accumulated over a sub-region and plotted as one-dimensional (1D) histograms, allowing easy comparison of the spectral characteristics of rainfall for the sub-region across temporal or spatial scales and between datasets.

### 2.1.2 Two-dimensional histograms

To diagnose the behavior of satellite-derived and simulated precipitation between consecutive temporal intervals at a fixed gridpoint, we construct two-dimensional (2D) histograms of gridpoint precipitation in temporal interval $t$ against precipitation at the same gridpoint in the next interval $t + \Delta t$, where $\Delta t$ is the sampling frequency of the input data. Gridpoint precipitation is binned, using bins that give a roughly uniform distribution for 2000–2012 TRMM analyses over an extended tropical Warm Pool domain (10°S–10°N, 60°–160°E), while also maintaining a pseudo-logarithmic scale. The 2D histograms are normalized by the total number of data points, such that the integral of the normalized histograms is unity. Figs. 2a,b show examples of this diagnostic for CMORPH 0.25° and 1.25° data. For a given cell $(i,j)$, the value shown is the joint probability of precipitation at a gridpoint in intensity bin $i$ during temporal interval $t$ and precipitation at the same gridpoint in intensity bin $j$ during temporal interval $t + \Delta$t. Averaging from 0.25° to 1.25° resolution slightly reduces the frequency of very heavy precipitation ($> 180$ mm day$^{-1}$) and near-zero precipitation, while slightly increasing the frequency of rates in between. Averaging also increases the probability of persistent precipitation in consecutive 3-hr intervals, as there are higher probabilities towards the central diagonal and lower probabilities along the axes in Fig. 2b relative to Fig. 2a.

### 2.1.3 Correlations with distance and lag

Correlations of precipitation in space and time indicate the typical scales of convective features. To compute these, we divide the analysis domain into non-overlapping sub-regions of 7×7 gridpoints. We select the central point in each region and extract the timeseries of precipitation. Computing the instantaneous correlation between the precipitation timeseries at each point in the sub-region and the central point, then averaging the resulting 7×7 correlation maps across all sub-regions in the analysis domain, creates a field of composite lag-0 correlations like those shown in Figs. 2c and 2d for CMORPH data. As expected, the correlations decrease with distance away from the central point. Correlations decrease more quickly along the diagonal axes, for which distances are greater, than along the major axes; correlations also decrease more quickly in the meridional direction than in the zonal direction, likely because the prevailing winds in our extended tropical Warm Pool domain are zonal. Correlations are lower for the 1.25° than for 0.25° CMORPH data, which is expected as each 1.25° gridpoint represents a 5x greater physical distance than at 0.25°.

While the composite correlation maps are useful, we are interested in both spatial and temporal scales of precipitation, which requires computing lagged correlations. It would be cumbersome to produce a set of composite correlation maps, one for each lag, for each datasets in this study. Instead, we developed a summary diagram that combines information about the spatial and temporal correlations of precipitation, based on the same 7×7 sub-regions. The construction of this diagram is described below; examples using CMORPH are shown in Figs. 2e and 2f.

We compute the distance ($d$; in km) between each point in the sub-region and the central point and convert this distance into units of $\Delta$x (the longitudinal grid spacing at the equator). We bin the gridpoints in the sub-region by their distance from the central point in $\Delta$x units, using bins of width $\Delta$x starting from 0.5$\Delta$x (e.g., 0.5$\Delta$x $< d \leq$ 1.5$\Delta$x, 1.5$\Delta$x $< d \leq$ 2.5$\Delta$x). We omit the bin of $0 < d \leq 0.5\Delta$x, as no datasets in our study have a grid with $\Delta$y $\leq 0.5\Delta$x in the tropics (where $\Delta$y is the latitudinal grid spacing). We treat the central gridpoint as a separate bin.

Within each distance bin, we compute the average correlation at a range of lags between the precipitation timeseries of gridpoints in that bin and the central gridpoint in the sub-region. For each 7×7 sub-region, these computations result in a matrix of correlations with distance and time, as shown in Figs. 2e and 2f. Note that all correlations are computed against the central point at lag=0. We average these matrices across all 7×7 sub-regions. For the central point (marked "Centre" in Figs. 2e and 2f), the result is simply the average of the autocorrelations of the central points in all sub-regions. At lag-0, the result is similar to the average of the correlations shown in Figs. 2c and 2d within each distance range. At all gridpoints, the precipitation timeseries is no longer statistically significantly correlated with itself (at $p$=0.05) after six hours (lag=2). At lag=2, the correlations at all distances are essentially uniform, including at the central point, which suggests that all spatial information from the lag=0 precipitation field has been lost (i.e., if the gridpoints in the lag=2 field were randomly swapped, one could not identify which was the central gridpoint). The CMORPH 1.25° data demonstrates that averaging increases correlations with time, due to the greater physical distance represented by each gridpoint (Fig. 2f), as significant correlations are maintained until nine hours (lag=3).

### 2.1.4 Comparisons among models and between models and analyses

The example of CMORPH 0.25° and 1.25° data demonstrates that the correlations in Figs. 2c–f are difficult to compare across datasets with different resolutions, because they are expressed as functions of the native gridscale. To compare spatial scales of precipitation features across resolutions, we repeat the method described in section 2.1.3 but using sub-regions defined by physical distance, rather than a number of gridpoints. To ensure that we include correlations to a distance of at least $2.5\Delta x$ in the coarsest-resolution models considered here (Table 1), and to optimize the number of sub-regions relative to the size of the domain, we divide the analysis domain into sub-regions of approximately $1500 \times 1500$ km, rounded to a distance equal to a whole number of gridpoints in the input dataset. Thus, the size of the sub-regions varies slightly from one dataset to the next. Within each sub-region, we bin the gridpoints by distance from the central gridpoint in units of $\Delta x$ and compute correlations as in section 2.1.3. A higher-resolution model or dataset will contain more gridpoints in each sub-region, and so have more distance bins, than a lower-resolution model or dataset, but this method allows a cleaner comparison between datasets of different resolutions.

For each dataset, we compute the minimum, median and maximum physical distance from the central point within each distance bin. This allows us to construct a graph of the correlation with physical distance at lag=0. Fig. 3a shows an example comparing TRMM and CMORPH at 0.25° and 1.25° resolutions. Each point represents one distance bin, plotted at the median distance for that bin; the horizontal solid lines span the minimum and maximum distance for that bin. Spatial averaging slightly increases correlations at the same distance for both TRMM and CMORPH. Estimates of the spatial scale of precipitation features from a finer-resolution dataset will be lower than those from a coarser-resolution version of the dataset.

To compare temporal correlations of precipitation, we use the mean auto-correlation of precipitation at all gridpoints within the analysis domain. Fig. 3b shows an example of this analysis, again for TRMM and CMORPH; each point represents one timestep in the input dataset. Spatial averaging increases estimates of the temporal scale of precipitation features.

We also create summary metrics of temporal and spatial coherence in precipitation. First, we compute quartiles of precipitation at each gridpoint, using only rates $> 1$ mm day$^{-1}$ to prevent near-zero precipitation values from dominating the lowest quartile. Computing quartiles separately at each gridpoint accounts for spatial variations in the distributions of precipitation across the analysis domain. To measure temporal coherence, we compute the probabilities that, at the same gridpoint and on consecutive timesteps, upper-quartile precipitation (U) is followed by upper-quartile precipitation [in probability notation, p(U|U)]; lower-quartile precipitation (L) is followed by lower-quartile precipitation [p(L|L)]; upper-quartile precipitation is followed by lower-quartile precipitation [p(L|U)]; and lower-quartile precipitation is followed by upper-quartile precipitation [p(U|L)]. When computing p(L|L) and p(L|U), the lowest quartile is expanded to include rates $\leq 1$ mm day$^{-1}$ for only the second timestep, to account for transitions to near-zero precipitation. In other words, p(L|U) is the probability that upper-quartile precipitation is followed by precipitation below the threshold for the lowest quartile, including rates $\leq 1$ mm day$^{-1}$. High values of p(U|U) and p(L|L) demonstrate temporal persistence; high values of p(L|U) and p(U|L) demonstrate temporal intermittency. As a metric of coherence ($M$), we combine these probabilities using

$$M = 0.5 \times [p(U|U) + p(L|L) - p(L|U) - p(U|L)] \tag{2}$$

High values of $M$ represent greater persistence. The factor of 0.5 ensures that the range of possible $M$ values spans -1.0 to 1.0. Negative $M$ indicates that intermittency is more common than persistence; positive values indicate that persistence is more common than intermittency.

To measure spatial coherence, we divide the analysis domain into non-overlapping regions of 3×3 gridpoints, in the same manner as for the 7×7 regions in section 2.1.3. In each region, we select the central gridpoint and find all instances of upper-quartile precipitation. For those timesteps, we compute the probability of upper-quartile precipitation at the eight other gridpoints in the 3×3 region [p(U|U)], as well as the probability of lower-quartile precipitation [p(L|U)]. We then compute similar probabilities for timesteps with lower-quartile precipitation at the central gridpoint [p(L|L) and p(U|L)]. As for the temporal persistence metric, we expand the lowest quartile to include values $< 1$ mm day$^{-1}$ when assessing precipitation at neighboring gridpoints. Finally, we compute $M$ using (2) above. As for temporal persistence, high values of $M$ represent greater spatial coherence, while low values represent spatial intermittency.

Table 3 shows that for TRMM and CMORPH, averaging from 0.25° to 1.25° horizontal resolution reduces the spatial coherence of precipitation, but increases the temporal persistence of precipitation. The reduction in spatial coherence is due to the metric being computed on the native gridscale of the input dataset, rather than physical distance; as above, the results in Fig. 3a show that the 1.25° datasets have larger spatial features than the 0.25° datasets. It is not practical to use physical distance in our summary spatial coherence metric, as selecting a fixed physical distance would be problematic for certain applications and regions (e.g., close to steep topography, land/sea contrasts).

### 2.1.5 Spatial and temporal averaging

To assess the sensitivity of sub-daily precipitation variability to the choice of spatial and temporal scale, we compute many of the above diagnostics using not only precipitation at a model's native gridscale and timestep, but also precipitation that has been averaged in time or space or both. For all models, we average timestep precipitation to 3-hr means for ease of comparison with TRMM and CMORPH. For all models and TRMM and CMORPH, we use an area-weighted method to average gridscale precipitation onto a common 5.6°×5.6° grid that is approximately four times coarser than the coarsest-resolution models used in this study. Using this grid, rather than the native grid of the coarsest-resolution models, ensures that all models are subject to some degree of spatial averaging, which our results show can substantially impact sub-daily precipitation statistics.

### 2.2 Models

We obtained gridpoint, timestep precipitation data from ten of the 12 models that participated in the two-day hindcast component of the "Vertical structure and physical processes of the Madden–Julian oscillation (MJO)" model-evaluation project (Xavier et al., 2015). The project was organised by the Global Atmospheric Systems Studies (GASS) panel, the Years of Tropical Convection (YoTC) and the MJO Task Force. We did not obtain data from the European Centre for Medium-range Weather Forecasts (ECMWF) Integrated Forecasting System, because ECMWF submitted hourly averages rather than timestep data. We omitted the Pacific Northwest National Laboratory configuration of the Weather Research and Forecasting model, because an incomplete dataset was archived. Table 1 lists the models, their timesteps and native horizontal resolutions, as well as ref-

erences with further details on their formulations. In the model-evaluation project, each model performed 48-hour hindcasts, initialized once per day from 00Z ECMWF operational analyses during two strong MJO events in boreal winter 2009–10. There are 22 start dates per event: 20 October–10 November 2009 and 20 December 2009–10 January 2010. To reduce the effects of model adjustment from the ECMWF analyses, we removed the first 12 hours of each hindcast, as in Xavier et al. (2015),

to leave 1584 hours of data (36 hours×44 hindcast dates) for each model. Data are available for all gridpoints in 10°S–10°N and 60°–160°E. Each of the two hindcast periods contains an active MJO that propagates from the Indian Ocean to the West Pacific, such that most gridpoints in the domain experience a transition from active to suppressed or suppressed to active MJO conditions during each event. This reduces the likelihood that our results depend upon MJO phase. TRMM and CMORPH are analysed at 1.25° resolution for the same period.

While the period of the hindcast experiments is relatively short, this is the only known multi-model dataset of timestep output from full-physics GCMs on the models' native grids. In addition, the dataset includes tendencies of temperature, humidity and winds from the individual sub-gridscale physical parameterizations in these models. While we do not consider these tendencies here, they represent a useful avenue for further research into the causes of the model behavior shown here.

For the GASS/YoTC models, TRMM and CMORPH, Table 2 gives the number of 7×7 sub-regions, the number of 1500×1500 km

sub-regions and the dimensions of the 1500×1500 km sub-regions in native gridpoints.

## 3    Results

In all figures, we order the GASS/YoTC models alphabetically by abbreviation (Table 1) except that we place MetUM-GA3 first. MetUM-GA3 often displays behaviour distinct from the other models. Because of the attention paid to MetUM-GA3 in our discussion, and because MetUM is the subject of our future work, we choose to separate this model to emphasize its unique

behaviour.

### 3.1    Behavior on the native grid and timestep

Two-dimensional histograms (section 2.1.2) reveal that the GASS/YoTC models vary considerably in their levels of temporal variability in gridpoint, timestep tropical precipitation (Fig. 4). On these diagrams, high probabilities along the central diagonal indicate persistent precipitation rates on consecutive timesteps at the same gridpoint. Low probabilities along the diagonal and

high probabilities in the lower-right and upper-left quadrants, close to the axes, identify intermittent precipitation at a gridpoint: high probabilities in the lower-right quadrant indicate that moderate or heavy precipitation is often followed by light or no precipitation, while high probabilities in the upper-left indicate that light or no precipitation is often followed by moderate or heavy precipitation. MetUM-GA3 has by far the most intermittent precipitation this measure. The 1D histogram suggests that MetUM-GA3 oscillates between lighter ($< 9$ mm day$^{-1}$) and heavier ($> 30$ mm day$^{-1}$) rain rates, with almost no instances of

moderate rates (9–30 mm day$^{-1}$). Heavier precipitation almost never persists for more than one timestep, while light or near-zero precipitation is much more likely to be followed by light or near-zero precipitation on the next timestep. This behavior suggests that when MetUM-GA3 triggers convection, if that convection is strong, the convection alters the thermodynamic

profile such that it is highly unlikely that strong convection will be triggered on the next timestep. The bi-modal 1D histogram suggests that most deep convection in MetUM-GA3 is as strong as possible, given the horizontal resolution and scientific configuration of the model.

Among the other models, CNRM-AM, GISS-E2, SPCAM3, ECEarth3 and CanCM4 show some degree of timestep inter-
mittency in precipitation. Unlike MetUM-GA3, however, all of these models have higher values on the central diagonal than away from it (i.e., the most likely value of precipitation at one gridpoint and timestep is the value of precipitation at the same gridpoint on the previous timestep). CNRM-AM and CanCM4 show behavior most similar to MetUM-GA3, with probabilities on the abscissa and ordinate axes that are nearly as high as those on the central diagonal. In CanCM4, the 2D PDF is almost uniform for rates $< 60$ mm day$^{-1}$, suggesting random behavior; rates $\geq 60$ mm day$^{-1}$ are more persistent.

In contrast, GEOS5, MRI-AGCM, CAM5 and MIROC5 are models with more persistent precipitation, in which gridpoint precipitation at one timestep is highly correlated with precipitation at the next timestep. These models maintain this behavior across the spectrum of intensity, such that even very heavy precipitation is much more likely to be followed by very heavy precipitation than by light or near-zero precipitation. This implies that in these models, strong convection does not result in a stable profile that inhibits convection on the next timestep. We note that there is no correspondence between the length
of the model timestep and temporal intermittency in precipitation: of the six models with 30-min timesteps (Table 1), three produce relatively intermittent precipitation (CNRM-AM, GISS-E2 and SPCAM3), while three produce relatively persistent precipitation (MRI-AGCM, CAM5 and MIROC5).

To evaluate spatial coherence of timestep precipitation and temporal variability at lags $> 1$ timestep, we use the diagnostic of the average correlation with distance and lag described in section 2.1.3 (Fig. 5). All models show decreasing correlations with
distance from the central point and with temporal lag, as expected. Despite having the finest horizontal resolution, MetUM-GA3 produces the lowest correlations of any model in space and time. All other aspects being equal, horizontal resolution should increase spatial correlations when measured as a function of $\Delta$x, as seen in Fig. 2 for CMORPH. The lag-1 correlation at the central gridpoint is slightly negative for MetUM-GA3. The correlation then increases at subsequent lags, reaching a maximum at lag-4. CNRM-AM also shows a lag-1 minimum in the auto-correlation of timestep precipitation, but the lag-1 correlation is
still strongly positive in that model (0.683). MetUM-GA3 also shows very low spatial coherence: the instantaneous correlation of precipitation at the central gridpoint with the precipitation at points 0.5–1.5$\Delta$x away is not statistically significant at the 10% level (r=0.13; p$\sim$0.15). This implies that timestep precipitation in MetUM-GA3 cannot be reliably predicted from precipitation at neighboring gridpoints at the same timestep, or from previous timesteps at the same gridpoint; it is quasi-random. CanCM4 displays similar behavior, with a instantaneous correlation of only 0.17 between the central point and points 0.5–1.5$\Delta$x away.
CanCM4 has a $\Delta$x that is approximately five times longer than MetUM-GA3, however. Indeed, with the exception of MetUM-GA3, models with coarser horizontal resolution (GISS-E2, SPCAM3, CanCM4) tend to show lower spatial correlations than models with finer resolution (GEOS5, CAM5, MRI-AGCM). This may be expected, since the physical area of the 7×7 boxes considered for this diagnostic will be far larger in the coarser-resolution models than in the finer-resolution ones. Naïvely, one would expect a larger area to have more spatially heterorogeneous large-scale forcing, and hence less coherent precipitation.
This hypothesis is difficult to confirm with such a wide variety of GCMs—which differ in many respects beyond horizontal

resolution (e.g., sub-gridscale parameterizations)—and suggests the need for resolution-based sensitivity experiments with a single model.

Fig. 6 compares the spatial and temporal scales of timestep precipitation in the GASS/YoTC models. On the native grid and timestep, MetUM-GA3 is clearly an outlier, with by far the lowest spatial (Fig. 6a) and temporal (Fig. 6b) coherence in precipitation. Only MetUM-GA3 and CNRM-AM show a lag-1 minimum in the auto-correlation of timestep precipitation; in MetUM-GA3 the correlation remains lower the other models for the remainder of the 3-hr period considered.

With the exception of MetUM-GA3, the models exhibit similar rates of decline in precipitation coherence with increasing distance. Models which show relatively higher correlations in first distance bin (CAM5, GEOS5, MIROC5, MRI-AGCM3) tend to have relatively higher correlations at longer distances; models with relatively lower correlations (CanCM4, CNRM-AM, SPCAM3) also maintain that behavior. The same is true for the decrease in correlation with increasing lag. There is a clear link between spatial and temporal coherence in these models: models which show relatively higher spatial coherence also tend to show relatively higher temporal coherence, and vice versa.

The summary metrics of coherence confirm these results (Table 3). The models identified as having the most temporally persistent precipitation based on 2D histograms and auto-correlations—CAM5, GEOS5, MRI-AGCM3 and MIROC5—also emerge as the models with the highest values of the temporal persistence summary metric. These models also show relatively high spatial coherence, as does ECEarth3. MetUM-GA3, CanCM4 and CNRM-AM display the least temporal persistence of precipitation, as well as low spatial coherence of precipitation; SPCAM3 also shows low spatial coherence of precipitation, but relatively higher temporal persistence of precipitation. MetUM-GA3 is the only model to produce a negative value for the spatial coherence metric, suggesting that spatial intermittency in precipitation is more common than spatial coherence.

To provide sample visualizations of the spatial and temporal character of precipitation in models with intermittent and persistent precipitation, we show example timeseries and maps of timestep, gridpoint precipitation from MetUM-GA3 and GEOS5 (Fig. 7). We selected MetUM-GA3 as our analysis suggests it is by far the model with the most intermittent precipitation; we selected GEOS5 because its timestep and horizontal resolution are similar to MetUM-GA3 (Table 1), but it produces more persistent precipitation in time and space (Table 3). Timeseries of precipitation at a gridpoint in the middle of the Indian Ocean ($0°$, $90°$E), for a forecast initialised during the active phase of the first MJO event (4 November 2010), confirms that MetUM-GA3 produces temporally intermittent precipitation (Fig. 7a). All timesteps with precipitation rates $> 5$ mm day$^{-1}$ also have rates $> 100$ mm day$^{-1}$. By contrast, GEOS5 produces some precipitation on nearly all timesteps, with only a few timesteps exceeding 100 mm day$^{-1}$ (Fig. 7b). Maps of instantaneous precipitation rates in the two models show that MetUM-GA3 precipitation is also intermittent in space, and dominated by gridpoints with precipitation rates $> 100$ mm day$^{-1}$ (Fig. 7c), while GEOS5 precipitation is far more continuous with a broader distribution of intensities (Fig. 7d).

Even after removing MetUM-GA3 as an outlier, it is obvious that the remaining models exhibit a broad range of spatial and temporal coherence in their precipitation features on the native grid and timestep. Next, we consider whether these timestep and gridpoint characteristics influence the models' behavior at on longer and larger scales.

## 3.2 Effects of temporal averaging

We begin by considering the impact of averaging from timestep to 3-hr data on the distributions of precipitation intensity in the GASS/YoTC models, using histograms of the fractional contribution from each of the precipitation bins defined in eq. 1 to the total precipitation (Fig. 8). As in Fig. 4, the timestep histograms demonstrate the range of precipitation intensities produced by these models, with MetUM-GA3 generating almost all of its precipitation from intense timestep events >100 mm day$^{-1}$ (Fig. 8a). Maps of contributions to the average precipitation rate confirm that this is true across most of the domain (Fig. 9a), not just in the regionally-aggregated statistics. Most of the other models produce the majority of their precipitation from 10–100 mm day$^{-1}$ timestep events, including ECEarth3, which favors the 10–50 mm day$^{-1}$ intensity range over most of the Warm Pool (Fig. 9b). There are no relationships between the preferred intensity of precipitation and timestep length or horizontal resolution.

When all data are averaged to a common 3-hr resolution, the differences between the models reduce considerably (Fig. 8b). While averaging barely affects the histogram for some models (CAM5, CNRM-AM, GEOS5, MIROC5), for other models averaging shifts the PDF considerably (CanCM4, MetUM-GA3, SPCAM3). For this latter set of models, the dominant effect is to reduce the contributions from heavy precipitation (>100 mm day$^{-1}$) and increase the contributions from moderate precipitation (10–50 mm day$^{-1}$). This is the expected result for averaging a random process, but it is not clear that timestep precipitation within a 3-hr window should be random. The effect is clearly greatest for MetUM-GA3, which has very low temporal coherence of timestep precipitation and a short timestep (i.e., more timesteps are averaged together to produce the 3-hr average). The models least affected by temporal averaging are those with persistent timestep precipitation rates (e.g., CAM5, MIROC5). All models produce a narrower histogram with a sharper peak for 3-hr means than for timestep data. Combined with the reduction in the inter-model spread with temporal averaging, the narrower histograms demonstrates that analysing only averaged precipitation hides a wide variety of model behavior at the timestep level.

For a model with temporally intermittent precipitation like MetUM-GA3, temporal averaging can have a powerful effect on conclusions about the dominant precipitation rate. MetUM-GA3 produces nearly all of its precipitation in timesteps with $\geq$ 100 mm day$^{-1}$ rates, but the right column of Fig. 9a demonstrates that if one analysed only 3-hr data, one would believe that tropical precipitation fell almost exclusively in 10–50 mm day$^{-1}$ events. This could have important implications for parameterization development. This issue does not affect a model with more persistent precipitation like ECEarth3, which at the timestep and 3-hr scale generates most of its precipitation from 10–50 mm day$^{-1}$ events (Fig. 9b).

While there are no observation-based constraints on timestep rainfall for similar spatial domains and temporal periods as the model data analysed here, at the 3-hr scale we compare gridpoint data from the models to 1.25° TRMM and CMORPH data. Both TRMM and CMORPH produce histograms that are broader than the models' histograms and which peak at heavier precipitation rates. This suggests that, over the relatively short hindcast period, all of the models produce their precipitation from too-frequent, too-light 3-hr events (Fig. 8b). However, as noted above, the model 3-hr histograms do not represent the full range of timestep precipitation rates.

Two-dimensional histograms of 3-hr data (Fig. 10) demonstrate that averaging reduces, but does not eliminate, the variations in temporal intermittency among the models seen in the timestep data. Models with higher temporal intermittency in timestep precipitation (e.g., MetUM-GA3, CNRM-AM, SPCAM, CanCM4) show reduced intermittency for 3-hr means, with higher values along the central diagonal and lower values along the axes. The reduced intermittency is particularly striking for MetUM-GA3, in which the bi-modal PDF of timestep precipitation (dashed line on Fig. 4a) becomes considerably more uniform. This implies that the frequent moderate 3-hr values (9–30 mm day$^{-1}$) arise from a linear combination of timesteps of near-zero and very heavy ($> 30$ mm day$^{-1}$) precipitation, since these moderate precipitation values are completely missing from the timestep PDF (Fig. 4a). This supports the results from the 1D histograms (Fig. 8). The reduced intermittency at the 3-hr scale may be most clear in MetUM-GA3 because the timestep intermittency was so strong, or because of the shorter timestep in MetUM-GA3 relative to the other models with intermittent precipitation, which increases the effect of the averaging because more timesteps are combined.

Conversely, models with more persistent timestep precipitation (e.g., GEOS5, MRI-AGCM, CAM5 and MIROC5) display increased intermittency when data are averaged to 3-hr means (compare Fig. 10 to Fig. 4). As for the models with more intermittent precipitation, this can be explained with a "regression to the mean" argument: averaging several timesteps of a model with less intermittent precipitation introduces variability into the 3-hr timeseries from the occasional deviation of the timestep precipitation away from the central diagonal in Fig. 4. These models show much smaller changes in the 1D histogram between the timestep and 3-hr scales, relative to models with more intermittent precipitation, which suggests that the 3-hr values arise from many timesteps with rates close to the 3-hr mean. Again, this supports the results from the 1D histograms. Table 3 confirms that temporal averaging on the native grid reduces inter-model variations in the temporal persistence summary metric, by increasing values for models with relatively low scores (e.g., MetUM-GA3, CanCM4, CNRM-AM) and reducing the values for models with relatively high scores (e.g., CAM5, MIROC5, MRI-AGCM3).

With the exception of MetUM-GA3, it is clear that models with longer timesteps tend to show greater intermittency in 3-hr precipitation. This is likely because in these models, fewer timesteps have been combined to create the 3-hr mean. Since sub-daily precipitation data (e.g., 3-hr means or timestep values sampled every 3-hr) are often used in studies of extreme events, such as tropical cyclones, it is worth noting this apparent correlation between model timestep length and variability in precipitation rates, which could introduce sampling uncertainty into these studies. We find no relationship between spatial resolution and temporal intermittency in 3-hr precipitation.

The 2D PDFs suggest that all models show greater persistence in 3-hr precipitation than TRMM (Fig. 10a) and CMORPH (Fig. 10b), which is confirmed by the temporal persistence metric (Table 3). SPCAM3, CNRM-AM, ECEarth3 and CanCM4 are perhaps closest to TRMM and CMORPH, but still produce more persistent precipitation than the satellite-derived analyses. The variations in spatial resolution among the models, and between the models and TRMM and CMORPH, make it difficult to compare the 2D PDFs and the summary metrics directly, however. Section 3.4 revisits the comparison between the models and the satellite-based analyses using precipitation data that has been interpolated to a common horizontal grid. We note that there are also differences between TRMM and CMORPH over this short period: CMORPH displays more frequent light precipitation than TRMM, although previous studies have shown that both products under-detect light rainfall (e.g., Huffman

et al., 2007; Tian et al., 2010); TRMM precipitation is less persistent than CMORPH (Table 3). Even given the uncertainty in the satellite-based analyses, however, all models show greater temporal persistence than the analyses.

Fig. 11 summarizes the impact of temporal averaging on the spatial scale of precipitation features. Averaging increases the spatial scale for all models, but most dramatically for MetUM-GA3, although that model still has relatively low spatial

correlations. When using 3-hr data, all models display higher correlations (greater coherence) than TRMM and CMORPH at distances shorter than 300 km, after which the TRMM and CMORPH correlations become statistically insignificant at the 5% level (r < ~0.2; Fig. 11b). CMORPH has slightly larger precipitation features than TRMM, as well as higher values of the spatial coherence summary metric (Table 3).

For model data, lagged correlations of 3-hr precipitation (i.e., as in Fig. 6b but for 3-hr data) over a 36-hr window were

dominated by the overly strong and regular diurnal cycle of precipitation in the models, which manifested itself in our diagnostics as a pronounced peak in the correlations at a 24-hr lag (not shown). TRMM and CMORPH displayed a much weaker and broader peak across lags of 18–30 hr, suggesting greater day-to-day variability in the timing of the diurnal maximum in tropical maximum in the satellite-observations than in the models.

### 3.3   Effect of spatial averaging

To investigate the effects of spatial averaging, we area-average timestep data from all models to a common $5.6° \times 5.6°$ (approximately 620 km) horizontal grid that is four times the resolution of the coarsest models (SPCAM and CanCM4). Spatial averaging increases timestep persistence of precipitation in all models, as shown by the 2D PDFs in Fig. 12 and the temporal persistence summary metrics in Table 3. As for averaging to 3-hr means, spatial averaging reduces intermittency most strongly in those models which either (a) have high levels of intermittency at the gridscale (e.g., MetUM-GA3, CNRM-AM, SPCAM3)

or (b) have finer native resolution, as more gridpoints are averaged to create each $5.6° \times 5.6°$ box (e.g., MetUM-GA3, GEOS5, CAM5, MIROC5). Both (a) and (b) apply to MetUM-GA3, so it is not surprising that spatial averaging substantially reduces temporal intermittency. At the $5.6°$ scale, MetUM-GA3 is still one of models with the most intermittent precipitation, but while it was an outlier at the gridpoint scale, it is now largely indistinguishable from the other models with intermittent precipitation (e.g., CanCM4, GISS-E2, SPCAM3). The other models with intermittent precipitation have a much coarser native

resolution than MetUM-GA3, however (Table 1), which means that those models have not "benefited" from combining as many gridpoints. For the purposes of these diagnostics, using a common horizontal grid or a common timescale does not necessarily create a fair comparison between models, due to differences in the number of points or timesteps, respectively, that are combined to create the average.

At the $5.6°$ scale, the 1D histograms of precipitation (dashed lines on Fig. 12) and the spectra of precipitation contributions

(Fig. 13a) become strikingly similar among the models, despite the variety of timestep lengths (12–60 min). This suggests that, when averaged over a broad enough region, these models produce similar spectra of timestep precipitation, even though the spectra of native-gridpoint precipitation varies considerably. For instance, the comparison of Fig. 4a and Fig. 12a suggests that within each $5.6° \times 5.6°$ region, MetUM-GA3 has only a few gridpoints with non-zero precipitation, but that those points show very heavy precipitation (e.g., 90–130 mm day$^{-1}$), as indicated in Fig. 8a. In MetUM-GA3, the difference between a $5.6° \times 5.6°$

region with relatively light (e.g., 5 mm day$^{-1}$) and relatively heavy (e.g., 30 mm day$^{-1}$) precipitation is likely that the latter region has a few more gridpoints with very heavy precipitation than the former. By contrast, the comparison of Fig. 4f and Fig. 12f, and the similarity of the MIROC5 spectra in Figs. 8a and 13a implies that MIROC5 has many precipitating gridpoints in each 5.6°×5.6° region, most of which have a precipitation rate similar to the average for the region. Models for which spatial

averaging results in little change in the 2D and 1D histograms are likely to have more spatially coherent precipitation, at least within a ∼5° region, than models for which spatial averaging causes large changes in the character of timestep precipitation.

Fig. 14 compares the impact of spatial averaging on the temporal scales of precipitation features across models. As for temporal averaging, spatial averaging increases the temporal scale of precipitation in all models, but most notably in models with intermittent precipitation such as MetUM-GA3 and CNRM-AM. Still, there is substantial inter-model spread in the

auto-correlations. Some models (CAM5, GEOS5, MIROC5, MRI-AGCM3) show nearly perfect correlations, while others (CanCM4, CNRM-AM, MetUM-GA3, SPCAM3) show relatively smaller values. Even when averaging fairly large (≈360,000 km$^2$) regions, the lag-1 minima in MetUM-GA3 and CNRM-AM remain, showing the strong effects of timestep intermittency from self-limiting convection in those models.

We do not show correlations with distance for the spatially averaged data, as those correlations rely on 1500 km×1500 km

sub-regions that contain only ≈4 5.6°×5.6° gridpoints. Larger sub-regions are not possible as the dataset spans only 20° latitude. However, this could be done for larger (e.g. global) datasets or for individual models with higher spatial resolution. Instead, we show our summary metrics for spatial coherence, which use regions of only 3×3 gridpoints (Table 3). Most models show similar coherence for timestep, 5.6°×5.6° resolution data; as for timestep, gridpoint precipitation, MIROC5 and CAM5 are the most coherent, with MetUM-GA3 and CanCM4 the least coherent. Note that it is not appropriate to compare the spatial

coherence metrics for the native-grid and 5.6°×5.6° data, since the metric is sensitive to the resolution of the input data.

### 3.4   Effect of spatial and temporal averaging

Combining spatial and temporal averaging produces the cleanest comparison possible among the models and between the models and TRMM and CMORPH, but at the expense of masking the timestep and gridpoint variability from Fig. 4. Histograms of precipitation intensity show that 3-hr averaging of the spatially-averaged data further reduces the differences between the

models' intensity spectra, as well as between the models and TRMM and CMORPH (Fig. 13). Most models still produce too-frequent precipitation at lighter rates than TRMM and CMORPH, even when analyzed on a common grid (Fig. 13c), a result which is emphasized by taking the difference between the models' spectra and the CMORPH spectrum (Fig. 13d). Differences between the models and TRMM are similar (not shown). All models except GISS-E2 generate too much of their precipitation from light events and too little from heavy events.

At the 5.6° and 3-hr scale, the models also produce remarkably similar levels of temporal persistence (Table 3), as well as highly similar precipitation PDFs (Fig. 15). All models show low levels of intermittency, with maxima in the 2D histogram along the central diagonal and minima along the ordinate and abscissa. The similarities are particularly notable given the wide variety of behavior seen at the timestep and gridpoint level. Even MetUM-GA3 produces a 2D histogram a precipitation PDF that agrees well with the other models. At these scales, the models also agree with TRMM and CMORPH, although CAM5,

GEOS5, MIROC5 and MRI-AGCM3 have slightly more persistent precipitation than the satellite-based analyses (Table 3). Most models show higher values of the spatial coherence metric than TRMM and CMORPH, suggesting that precipitation features are still too broad.

The convergence of model behavior at the $\sim$600 km, 3-hr scale, combined with the close agreement with TRMM and CMORPH, implies a natural compensation in these models at the gridpoint and timestep level between the spatial and temporal intermittency in precipitation and the precipitation PDF. In other words, it seems that the models "adjust" the frequency and intensity of precipitation at their native resolutions to maintain an appropriate distribution of tropical precipitation at the broader $\sim$600 km and 3-hr scales. We hypothesize that these broader scales represent those at which simulated convection is in balance with the synoptic-scale, dynamical systems that produce precipitation, predictions of which should be highly similar among the models in the short, 2-day hindcasts we analysed. At finer and shorter scales, the models have sufficient degrees of freedom to produce the broad spectrum of behavior seen in Fig. 4 and Fig. 5. Therefore, it appears that the nature of the timestep, gridpoint variability does not substantially affect the distribution of precipitation or its variability at the $\sim$600 km and 3-hr scales. However, it remains unclear whether a model's timestep, gridpoint behavior influences other aspects of the simulation (e.g., through interactions between convective heating and the resolved dynamics). We discuss this further in section 4.

## 4   Discussion

Our diagnostics reveal that analyzing temporally or spatially averaged precipitation can hide a wealth of information about model behavior on the native gridscale and timestep. This is true even for relatively small averaging scales, such as 3-hr means or 2×2 gridboxes (our 5.6°× 5.6° regions were 4× the gridscale of the coarsest resolution models in our dataset). Analysis of gridpoint, timestep precipitation is critical for developing sub-gridscale parameterizations, since these are the scales at which the parameterizations interact with the resolved dynamics. Such analysis can identify potentially undesirable characteristics, such as the strong spatial and temporal intermittency in convection in MetUM-GA3. Nearly all of the convection in MetUM-GA3 is very strong, producing precipitation rates >100 mm day$^{-1}$ on a timestep (Fig. 8a); also, convection is often isolated to a single gridpoint and timestep (Fig. 5a). Although there are no verifying observations for the model timestep data that cover comparable spatial and temporal domains, it is difficult to believe that this behavior is representative of oceanic tropical convection. These intense, isolated precipitation features must be associated with intense, isolated column heating. Over a sequence of timesteps, this behavior produces a "checkerboard"-style spatial pattern of heating that shifts from one timestep to the next as gridpoint convection triggers quasi-randomly. It is not clear whether the model dynamics respond to this strong gridscale heating, or only to the average heating over several gridpoints and timesteps, but gravity waves triggered by the intermittent heating in one column may influence the likelihood of convection at neighboring gridpoints on subsequent timesteps, disrupting convective organization and the propagation of waves with longer periods and larger horizontal scales (e.g., Kelvin waves, the MJO). Understanding the controls on spatial and temporal intermittency in MetUM convection, as well as the influences of that intermittency on the model dynamics, tropical convective variability and the mean state, are all active areas of research inspired by our diagnostics.

Future research should also seek to validate timestep model convection against high-resolution observations, through comparison of model data against ground-based or space-borne precipitation radar measurements (e.g., from the Global Precipitation Measurement mission). These comparisons must take care to analyse observed and simulated data at comparable spatial and temporal scales, given our results on the effects of spatial and temporal averaging on the distributions and coherence of precipitation. Model development efforts to reduce or remove undesirable intermittency may involve single-column model experiments, in which the effects of changes in sub-gridscale physics can be isolated from feedbacks through the resolved dynamics (e.g., Satoh and Hayashi, 1992; Takata and Noda, 1997; Woolnough et al., 2010), although we stress that physics–dynamics coupling may have a substantial effect on the model behaviors and diagnostics presented here.

Although MetUM-GA3 is the model with the most intermittent precipitation in our study, CNRM-AM, CanCM4, GISS-E2, ECEarth3 and SPCAM3 display varying degrees of intermittency (Fig. 4). It is likely that all of those models have a self-limiting character to their convective parameterizations, such that the effect of convection on one timestep reduces the probability of convective for one or several subsequent timesteps. Preliminary analysis of MetUM-GA3 (not shown) suggests that convection on one timestep produces downdraft cooling that stabilizes the vertical temperature profile near the lifting condensation level (LCL), the stability across which is used in the diagnosis of deep convection (i.e., to diagnose deep convection, the parcel must be able to ascend through the LCL). Although instability may remain aloft, the model is unable to convect on subsequent timesteps until the profile again becomes unstable at the LCL. There are a variety of mechanisms by which a parameterization can be self-limiting, which will depend on the precise design of the parameterization; a detailed examination of the convective parameterizations of ten GCMs is outside the scope of this study, but our analysis of this behavior may be of interest to individual modeling centers to understand and improve their parameterizations.

On the gridpoint and timestep scale, the worlds simulated by these models are definitely not "dreary" (e.g., Stephens et al., 2010), at least over the Warm Pool domain considered here. In most models, the total precipitation consists of a variety of timestep rates that span 1–100 mm day$^{-1}$, with most precipitation falling in timesteps with precipitation rates $> 10$ mm day$^{-1}$ (Fig. 8a). Only when the timestep data are averaged to 3-hr means do the precipitation spectra begin to collapse to be lighter (Fig. 8b) and more persistent (Fig. 10) than in the satellite-derived analyses. The narrower spectra arise from the tendency for one timestep with heavy precipitation to be followed by several timesteps with no precipitation; the persistence of 3-hr rain rates suggests that the timestep intermittency occurs consistently in each 3-hr window. These results imply that the self-limiting character of a model's convection, displayed through temporal intermittency in timestep precipitation, prevents the model from producing enough consecutive timesteps of heavy precipitation, or enough consecutive timesteps of no precipitation, to generate a broader distribution of 3-hr mean rates. An observer stationed on an island in the Warm Pool in many of these models would be constantly dodging intense, short-lived downpours, not standing in the persistent light rain implied by past studies' analysis of 3-hr or daily mean data.

Much of our analysis has focused on timestep and gridpoint data from GCMs, the formulations of which include spatial and temporal smoothing (either implicitly or explicitly), as well as truncation errors, both of which lead to an underestimation of energy on the smallest resolved scales. Previous studies have found that the "effective resolution" of a GCM—the scales at which the truncation and smoothing have no effect, or zero power—is several times the native resolution (e.g., Skamarock,

2004; Frehlich and Sharman, 2008; Larsén et al., 2012), such that the timestep, gridpoint data are unreliable and should be discarded. While we do not argue with the conclusions of those studies, we believe that it remains important to examine the characteristics of native-resolution data for several reasons: (a) to inform parameterization development, as discussed above; (b) to understand the effects of intermittency on these scales, however under-resolved, because that intermittency may influence the larger and longer scales in a GCM; and (c) because despite previous conclusions on effective resolution, the scientific community is increasingly using gridscale, instantaneous output from models with ever-finer horizontal resolution to study extreme events and their responses to natural variability and anthropogenic climate change (e.g., Kendon et al., 2014).

We used 2-day hindcasts from the "Vertical structure and physical processes of the MJO" model-evaluation project, which is the only known source of timestep, gridpoint precipitation data from many contemporary models. However, this dataset has limitations. First, only two sets of 22 2-day forecasts were performed, each for a case study of an MJO event in boreal winter 2009–10. Although each set of forecasts samples the MJO active and suppressed phase, limiting the possibility of sensitivity to MJO phase, there is an active MJO in the analysis domain throughout the dataset, which may bias the simulated precipitation characteristics. We plan to address this issue in a future study by computing our diagnostics for across an entire season of MetUM timestep data. Secondly, the spatial domain of the data is limited to the deep tropical Warm Pool; the dataset may not represent the full spectrum of tropical convection in the models or satellite-derived analyses. Thirdly, all forecasts were initialized from ECMWF analyses. Xavier et al. (2015) found this led to an initialization shock, the strength of which varied among the models. To reduce the effect of the shock, we removed the first 12 hr of each forecast, as in Xavier et al. (2015), but it is possible that our findings are influenced by the shock and may not represent the model's intrinsic behavior. Removing the first 24 hr of each forecast made only a very small difference to our results and did not affect our conclusions.

The analysis in section 3 is only one potential use of these diagnostics. Understanding the spatial and temporal characteristics of precipitation is important for a variety of applications. Computing precipitation spectra (Fig. 8) and 2D histograms (Fig. 4) for daily-mean or pentad-mean precipitation from models and observations could give insight into the simulated levels of synoptic and intraseasonal variance in a particular region, for instance the active and break periods of the major monsoons. Spatial maps of contributions from sections of the precipitation spectra (Fig. 9) could aid understanding of whether biases in simulated mean precipitation are due primarily to biases in frequency or in intensity. Spatial and temporal coherence diagnostics (Fig. 3) may provide information on convective aggregation, which is important for tropical cyclones and the MJO. All of these diagnostics could be used to compare precipitation characteristics from simulations of the same model at various horizontal resolutions, or with perturbations to one or several parameters, to assist model development and assessment. We believe that these diagnostics will be useful primarily on sub-monthly and sub-2000 km scales, as larger and longer scales are likely dominated by the seasonal cycle rather than the individual synoptic or mesoscale systems that produce precipitation.

When comparing datasets with different spatial and temporal resolutions, it is commonplace to average all data to the resolution of the coarsest dataset. However, our results show that any spatial or temporal averaging can alter precipitation characteristics, such that for the purposes of these diagnostics it is unfair to compare a lower-resolution dataset at its native resolution to a higher-resolution dataset that has been averaged to the lower resolution. Instead, we recommend comparing the datasets at their native resolutions—to understand the behavior of each dataset—as well as at a common resolution at least 2×

(in each direction) that of the coarsest dataset in space and time. This is still not a clean comparison because the effects of averaging increase with the number of points combined (up to some asymptotic limit), but at least it allows both datasets to "experience" some averaging in space and time.

## 5 Conclusions

We have developed a range of diagnostics to identify the spatial and temporal characteristics of precipitation in observations and GCMs; these diagnostics form a small software package, "Analyzing Scales of Precipitation" version 1.0 (ASoP1). The ASoP1 diagnostics are designed be applied to sub-monthly data at horizontal resolutions O(1000 km) or finer, to assess precipitation variability associated with phenomena ranging from individual cloud systems to mesoscale weather systems and synoptic fronts. The diagnostics are motivated by the increasing attention paid to the simulation of local and regional hydrological extremes in fine-resolution GCMs—which often requires gridscale, instantaneous precipitation data—while model evaluation has remained focused primarily on monthly and seasonal accumulations. Sub-gridscale parameterization development requires information about the spatial and temporal variability of precipitation at the native gridscale and timestep, since these are the scales at which the parameterizations operate. The ASoP1 diagnostics include 1D histograms and spatial maps of the contributions of intensity ranges to the total precipitation (e.g., Fig. 8 and Fig. 9); 2D histograms of precipitation rates at the same gridpoint on consecutive time intervals (e.g., Fig. 2a), which show the temporal persistence of precipitation; the average correlation of precipitation at a range of distances and temporal lags, correlated against precipitation at a central gridpoint (Fig. 2c), computed by dividing the analysis domain into a series of non-overlapping sub-regions (e.g., Fig. 2b); average correlations as a function of either physical distance (in km) or time, with which one can compare datasets with different spatial and temporal resolutions (e.g., Fig. 3); and summary metrics that can be used to track easily the effects of changes to model resolution, physics or dynamics on the spatial and temporal coherence of precipitation (Table 3).

To demonstrate the value of these diagnostics, we apply them to ten models from the "Vertical structure and physical processes of the MJO" model-evaluation project (Table 1), which collected timestep data at the native model horizontal resolution over an extended Warm Pool domain (10°S–10°N, 60°–160°E) from 44 2-day hindcasts during two strong MJO events in boreal winter 2009–10. At the timestep and gridscale, some models produce precipitation features that are highly coherent in space and time, while others produce intermittent precipitation that resembles uncorrelated noise (Fig. 4). MetUM-GA3 is the model with the most intermittent precipitation, with a weakly negative lag-1 auto-correlation of timestep precipitation and no statistically significant correlations between precipitation at neighboring gridpoints (Fig. 6). We found no relationship between the level of intermittency and either horizontal resolution or the length of the model timestep. Models with intermittent precipitation tend to produce more of their total precipitation from very heavy events—often exceeding 100 mm day$^{-1}$ in the case of MetUM-GA3—while models with persistent timestep precipitation, such as ECEarth3, generate more frequent precipitation with moderate intensities of 10–50 mm day$^{-1}$ (Figs. 8 and 9). Strong and highly intermittent convection, such as that in MetUM-GA3, will be associated with strong and intermittent column heating, which may interact with the resolved dynamics, affecting the spectrum of tropical wave activity and even the mean state. The effects of this intermittency remain

unclear, but are an active area of research. The fact that five of the ten GCMs in this study produce heavy timestep precipitation rates, interspersed by timesteps of little or no precipitation, contradicts the common criticism that GCMs simulate a "dreary state" in the tropics of continual light precipitation, which arose from studies that analyzed 3-hr or daily averaged precipitation (e.g., Stephens et al., 2010). In fact, many models continually produce short-lived, intense downpours throughout the Warm Pool.

Averaging timestep, gridscale data in either time (to 3-hr means) or space (to 5.6°×5.6°) considerably reduces inter-model variations in the spatial and temporal scales of precipitation (Figs. 11 and 14, Table 3), as well as in the spectra of precipitation intensities (Fig. 8) and the temporal persistence of precipitation rates (Figs. 10 and 12). This is because spatial or temporal averaging has a greater effect on intermittent precipitation than on persistent precipitation. When compared to TRMM and CMORPH satellite-derived precipitation analyses over the same period and domain, all models produce 3-hr precipitation features that are too broad and too persistent, despite the fact that many of those same models produce timestep precipitation features that are isolated in both space and time (Fig. 11). This emphasizes that averaging in either space or time can hide a wealth of information about the intrinsic behavior of GCMs.

Averaging 3-hr data from the models, TRMM and CMORPH to a common 5.6°×5.6° grid improves the agreement among the models, as well as between the models and the satellite-derived analyses (Figs. 13 and 15, Table 3). We hypothesize that the strong agreement among the models indicates that these are the scales at which the convection in these models is in balance with the synoptic-scale, dynamical systems that produce precipitation. This convergence of model behavior may be enhanced by the fact that these data are from short (2-day) forecasts initialized from the same ECMWF analyses, which means that the representation of these dynamical systems are much more similar among models than if the data came from free-running climate simulations.

These results represent only one possible use of the ASoP1 diagnostics, which we believe will be useful for model development and evaluation at longer (e.g., daily, synoptic) and larger (e.g., regional averages) scales, as well as at the native gridpoint and timestep. In particular, these diagnostics would be ideal for understanding the effects of horizontal resolution and changes to physical parameters on the simulated spatial and temporal scales of precipitation, and for comparing the characteristics of precipitation and their representation in models in different tropical regions.

## 6   Code availability and requirements

The ASoP1 diagnostics package is coded in Python 2. The code is available under the terms of the Apache 2.0 license from the lead author's GitHub repository at https://github.com/nick-klingaman/ASoP. There are two software packages: ASoP-Spectral, which computes the 1D histograms and maps of the contributions of specific intensity bins to the total precipitation; and ASoP-Coherence, which computes the 2D PDFs, the correlations with distance and time and the spatial and temporal coherence summary metrics. The user must install several Python packages prior to running the code; a list of these is given at the top of each python code file in the package. These packages also have software dependencies. The hardware requirements for running

the code will vary based on the size of the dataset the user wishes to analyze, particularly for the amount of system memory (RAM) required. The analysis shown in this manuscript was performed on a four-core desktop workstation with 32GB RAM.

## 7 Data availability

Data from the "Vertical structure and physical processes of the Madden–Julian oscillation" project can be obtained from the Earth System Grid Federation: https://www.earthsystemcog.org/projects/gass-yotc-mip.

TRMM 3B42 version 7A data can be obtained from http://disc.sci.gsfc.nasa.gov/TRMM.

CMORPH version 1.0 data can be obtained from ftp://ftp.cpc.ncep.noaa.gov/precip/global_CMORPH/3-hourly_025deg.

*Author contributions.* N. Klingaman developed the diagnostics using 2D histograms, correlations versus distance and lag and correlations in space and time. G. Martin and A. Moise developed the diagnostics using 1D histograms and maps of the contributions of intensity bins. N. Klingaman wrote the manuscript with input from all co-authors.

*Acknowledgements.* N. Klingaman was supported by an Independent Research Fellowship from the UK Natural Environment Research Council (NE/L010976/1). G. Martin was supported by the Joint DECC/Defra Met Office Hadley Centre Climate Programme (GA01101).

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

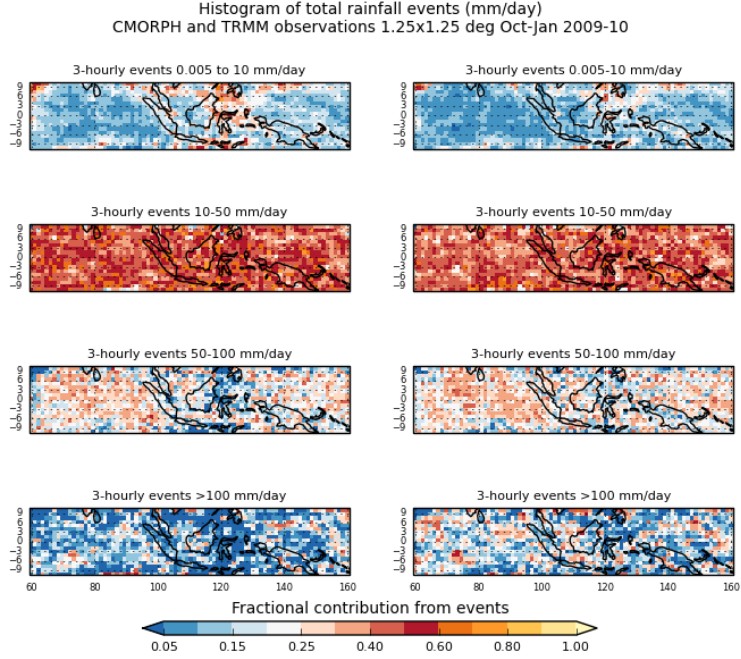

**Figure 1.** For (left) CMORPH and (right) TRMM 3B42 1.25° data, the fractional contribution to the total precipitation rate from ranges of intensity bins shown in the labels above each panel. For each dataset, the sum of each column is unity.

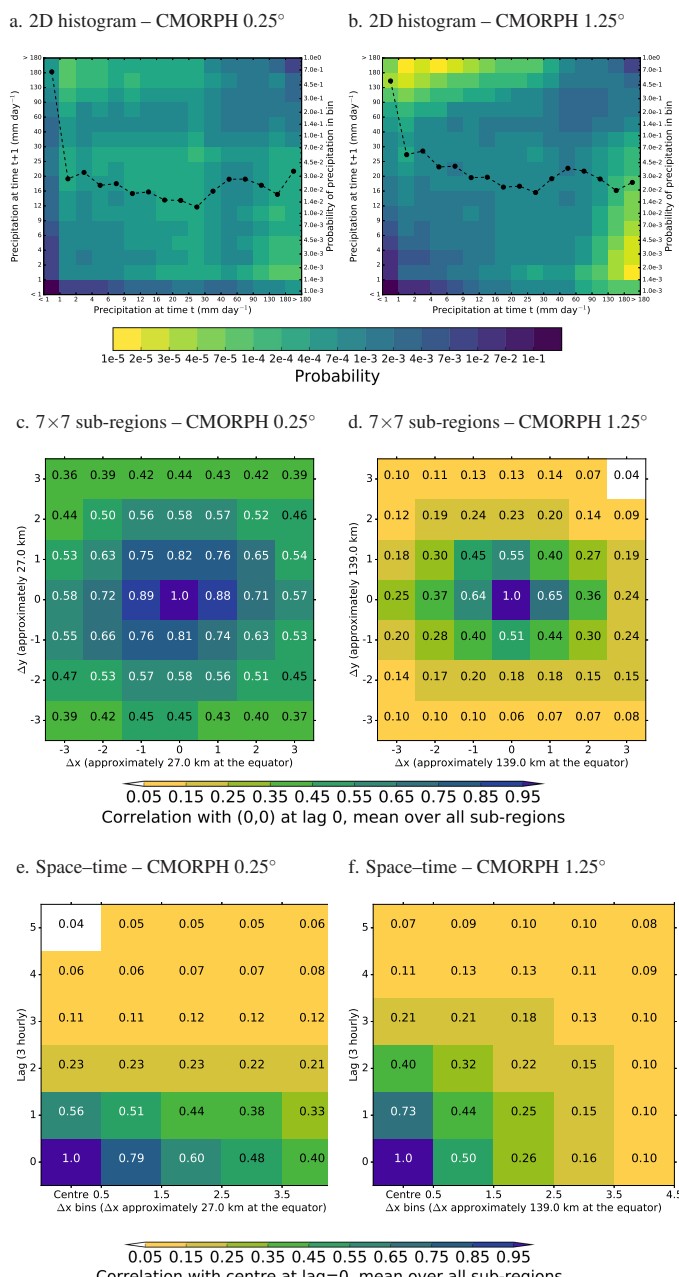

**Figure 2.** For CMORPH (a,c,e) 0.25° and (b,d,f) 1.25° data: (a,b) filled blocks show the 2D histogram of binned values on consecutive 3-hr steps at the same gridpoint, aggregated over all gridpoints; the dashed line shows the 1D histogram, using the right-hand axis; (c,d) the lag-0 correlation between each gridpoint in a 7×7 region and the central gridpoint (0,0), averaged over all non-overlapping 7×7 gridpoint regions in the domain; (e,f) lagged correlations between the central gridpoint in each 7×7 region and gridpoints within each range of distance on the horizontal axis away from the central point, averaged over all 7×7 regions. In (a,b), note the logarithmic color scale; in (c–f), the printed values and filled blocks show the same data. In (e,f) we omit the bin for points less than 0.5Δx away from the central point, as no points in these datasets fall into that bin; "centre" is the auto-correlation at the central point.

## a. Correlations with distance

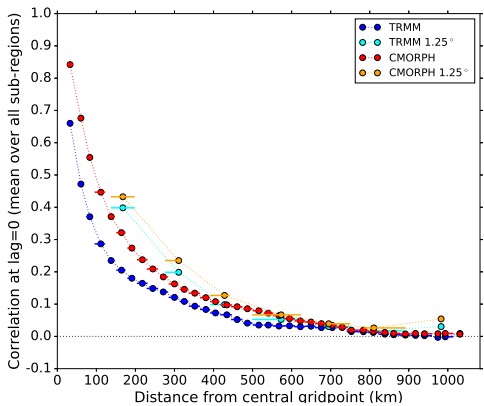

## b. Correlations with time

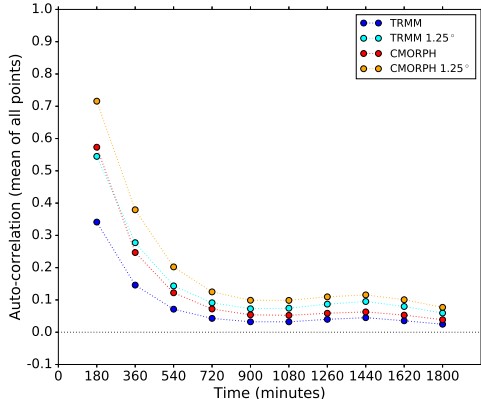

**Figure 3.** For TRMM 3B42 and CMORPH 0.25° and 1.25° data: (a) a measure of the spatial scale of precipitation features, computed by dividing the analysis domain into 1500×1500 km regions and calculating the lag-0 correlation between the central gridpoint and gridpoints within each distance bin (which are $\Delta$x wide, starting from $0.5\Delta$x) away from the central gridpoint, then averaging the correlations over all regions in the domain; (b) a measure of the temporal scale of precipitation features, computed as the auto-correlation of precipitation, averaged over all points in the domain. The horizontal lines in (a) show the range of distances spanned by each distance bin; the filled circle is placed at the median distance. For clarity, we omit the correlations for zero distance and zero lag, which are 1.0 by definition.

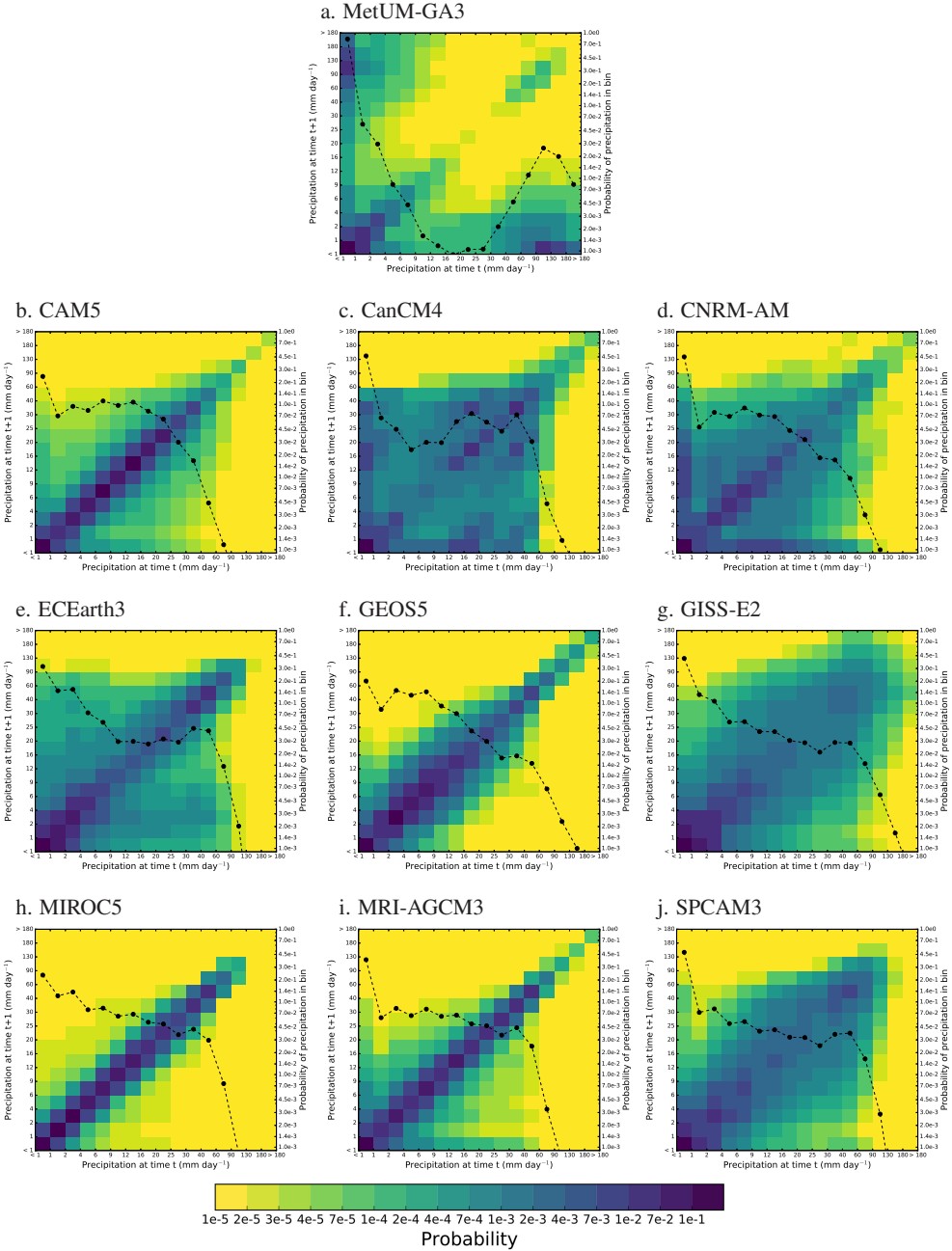

**Figure 4.** For each model in the GASS/YoTC dataset and using timestep precipitation data on the native model grid: filled blocks show the normalized 2D histogram of binned values on consecutive timesteps, aggregated over all gridpoints; the dashed black line shows the normalized 1D histogram, using the right-hand axis. Note the logarithmic color scale. See Table 1 for information on timestep length and grid spacing.

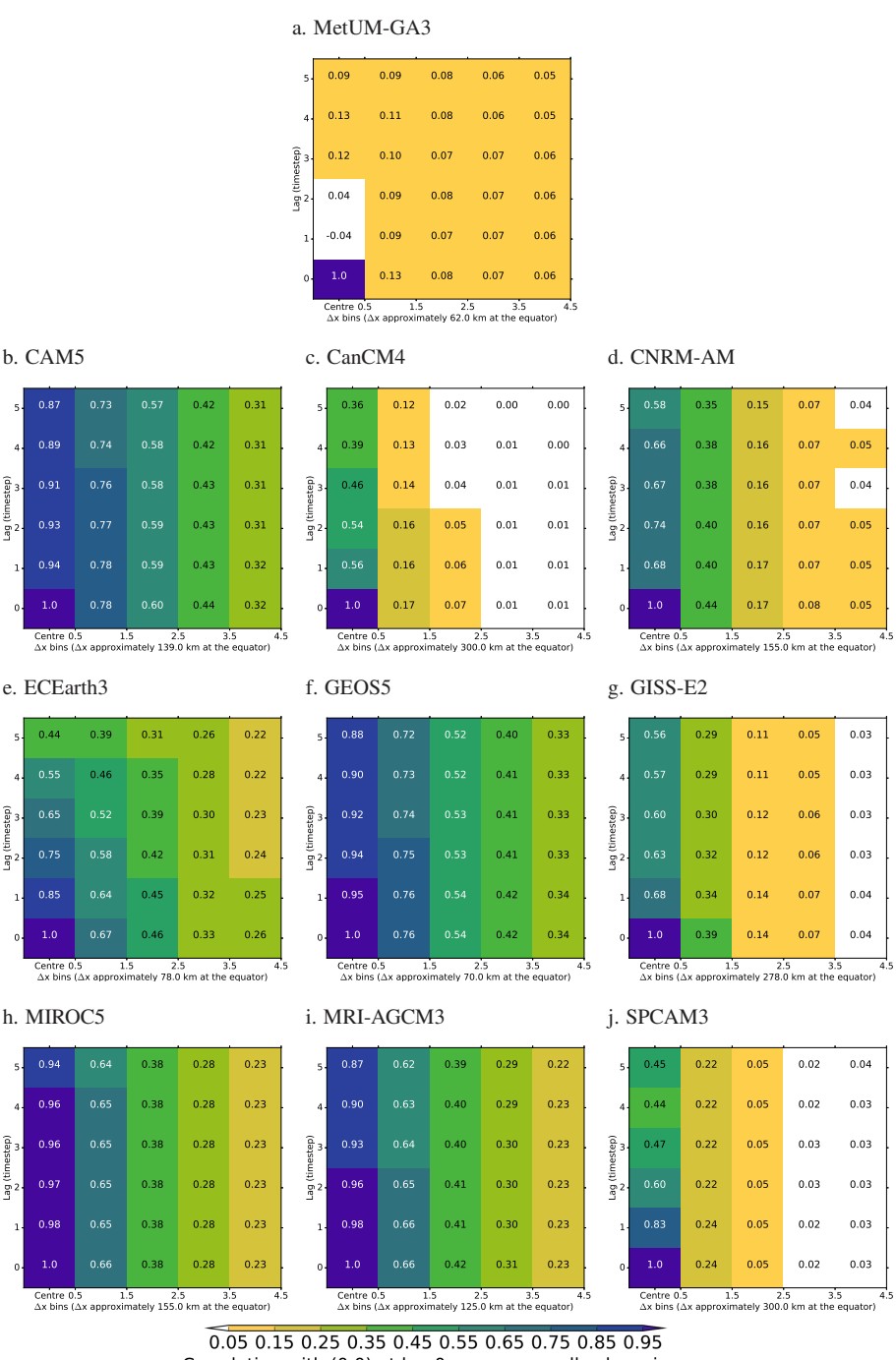

**Figure 5.** For each model in the GASS/YoTC dataset and using timestep precipitation data on the native model grid, lagged correlations between the central gridpoint in each $7{\times}7$ region and gridpoints within each range of distance on the horizontal axis (in units of $\Delta$x) away from the central point, averaged over all $7{\times}7$ regions. The printed values and filled blocks show the same data; "centre" in the auto-correlation at the central point. We omit the bin for points less than $0.5\Delta$x away from the central point, as no points in these datasets fall into that bin.

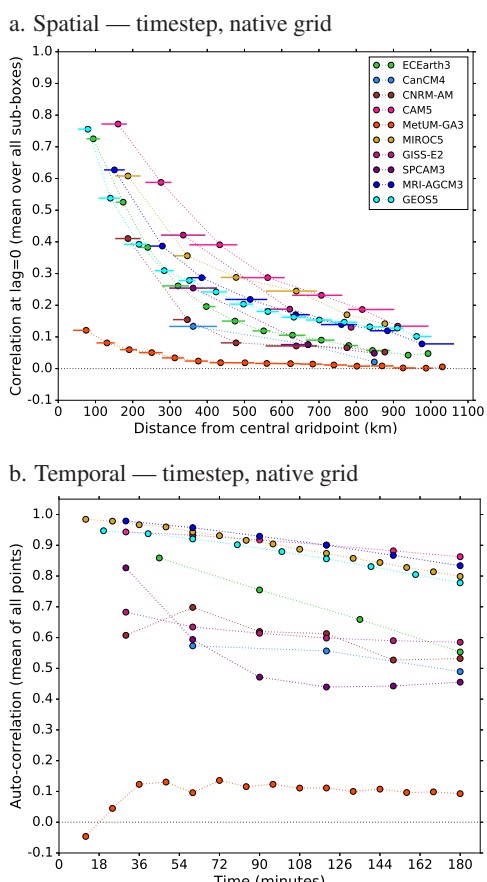

**Figure 6.** For the GASS/YoTC models, using timestep precipitation data on the native model grid: (a) a measure of the spatial scale of precipitation features, computed by dividing the analysis domain into $1500 \times 1500$ km regions and calculating the instantaneous linear correlation between the central gridpoint and gridpoints within each distance bin (which are $\Delta$x wide, starting from $0.5\Delta$x) away from the central gridpoint, then averaging the correlations over all regions in the domain; (b) a measure of the temporal scale of precipitation features, computed as the auto-correlation of precipitation, averaged over all points in the domain. The horizontal lines in (a) show the range of distances spanned by each distance bin; the filled circle is placed at the median distance. For clarity, we omit the correlations for zero distance and zero lag, which are 1.0 by definition.

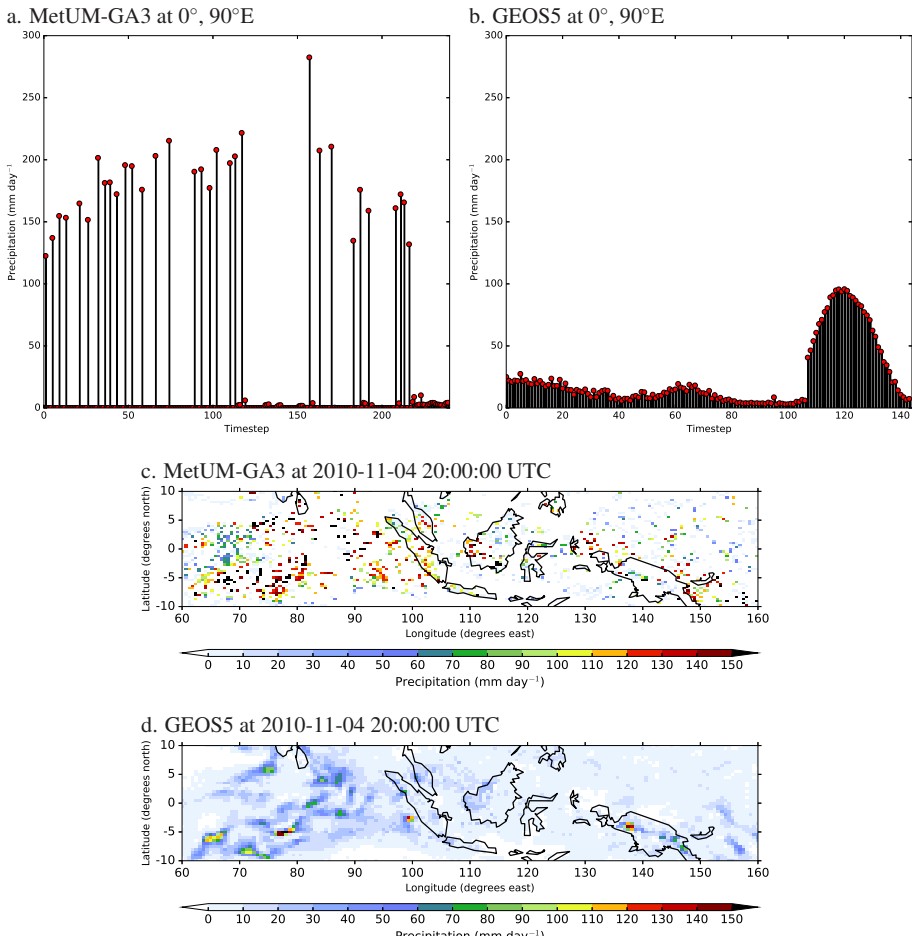

**Figure 7.** For (a) MetUM-GA3 and (b) GEOS5, timeseries of timestep, gridpoint precipitation (mm day$^{-1}$) at the point 0°, 90°E for the 48-hour forecast initialized at 00:00 UTC 4 November 2010. Each red dot represents one timestep; a red dot on the horizontal axis corresponds to zero precipitation. The series covers 48 hours for both models, which corresponds to 240 timesteps in MetUM-GA3 and 144 timesteps in GEOS5; the width of the vertical bars has been scaled for the difference in timestep length. For (c) MetUM-GA3 and (d) GEOS5, maps of instantaneous precipitation rates at the timestep corresponding to 20:00 UTC on 4 November 2010, from the forecast initialised on 00:00 UTC on 4 November 2010.

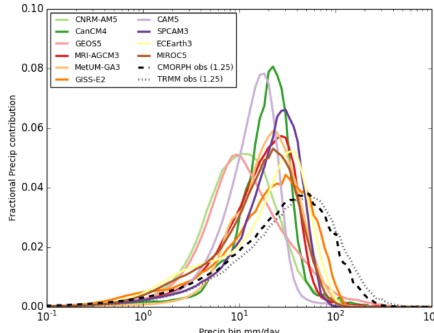

**Figure 8.** For the GASS/YoTC models, histograms of the fractional contributions from each precipitation bin (defined in eq. 1) to the total precipitation, computed across 60°E–160°E, 10°S–10°N, using data on the native horizontal grid and (a) timestep and (b) 3-hr averages. Panel (b) includes TRMM 3B42 and CMORPH at 1.25° resolution for the same region and time period as the GASS/YoTC models.

**Table 1.** For each model from the "Vertical structure and physical processes of the Madden–Julian oscillation" project from which timestep rain rates are used: the model name, the institution that produced the data, the horizontal resolution at the equator in degrees (to the nearest 0.01°) and the equivalent in km, the model timestep (Δt) in minutes and a reference with further details. Models are ordered alphabetically by abbreviation.

| Model name | Abbreviation | Lon°×Lat° (km) | Δt | Reference |
|---|---|---|---|---|
| CAM[1] | CAM5 | 1.25×0.94 (139×118) | 30 | Neale et al. (2012) |
| Canadian Coupled Model | CanCM4 | 2.80×2.80 (311×311) | 60 | Merryfield et al. (2013) |
| CNRM-AM[2] | CNRM-AM | 1.40×1.40 (155×155) | 30 | Voldoire et al. (2013) |
| European Community Model | ECEarth3 | 0.70×0.70 (78×78) | 45 | Hazeleger et al. (2012) |
| GEOS[3] | GEOS5 | 0.63×0.50 (70×55) | 20 | Rienecker et al. (2008) |
| GISS[4] GCM | GISS-E2 | 2.50×2.50 (278×278) | 30 | Schmidt et al. (2014) |
| Met Office Unified Model | MetUM-GA3 | 0.56×0.38 (62×42) | 12 | Walters et al. (2011) |
| MIROC[5] | MIROC5 | 1.40×1.40 (155×155) | 30 | Watanabe et al. (2010) |
| MRI[6] Atmospheric GCM | MRI-AGCM3 | 1.13×1.13 (125×125) | 30 | Yukimoto et al. (2012) |
| Super-Parameterized CAM | SPCAM3 | 2.80×2.80 (311×311) | 30 | Khairoutdinov et al. (2008) |

[1] Community Atmospheric Model

[2] Centre National de Recherches Météorologiques Atmospheric Model

[3] Goddard Earth Observing System

[4] Goddard Institute for Space Studies

[5] Model for Interdisciplinary Research on Climate

[6] Meteorological Research Institute

## a. MetUM-GA3

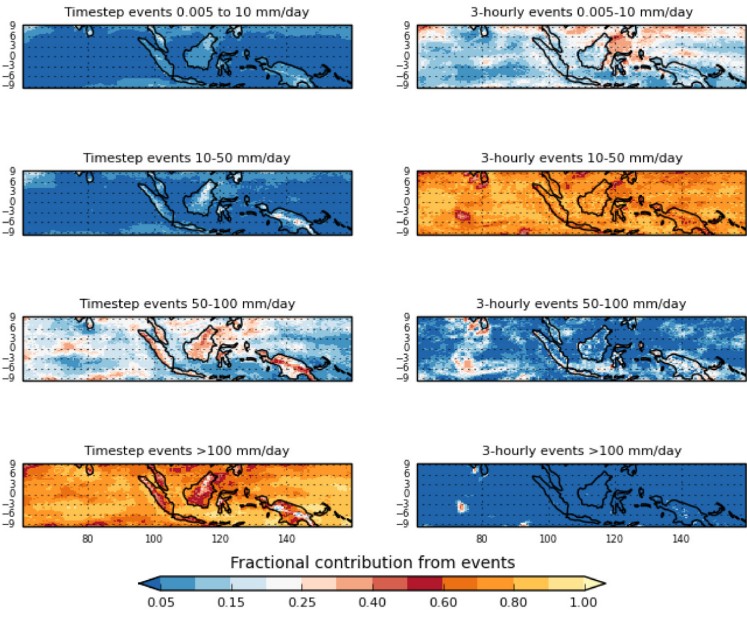

## b. ECEarth3

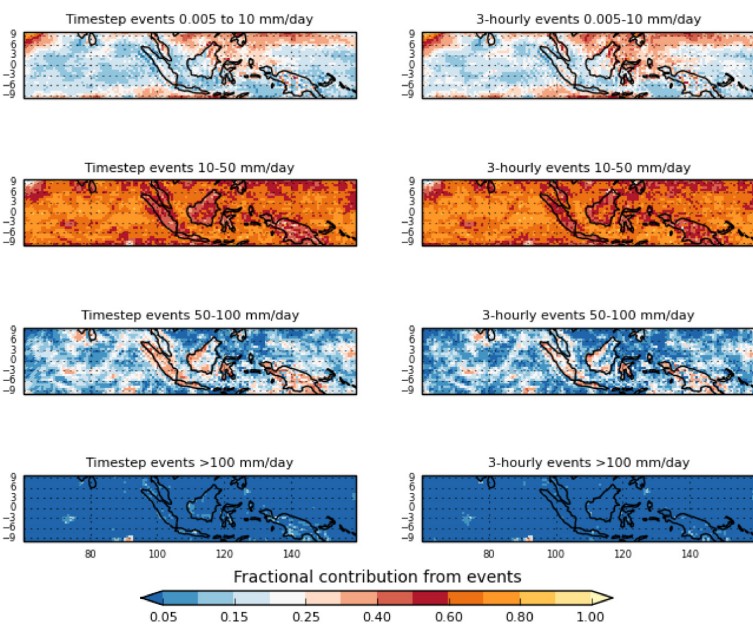

**Figure 9.** For (a) MetUM-GA3 and (b) ECEarth3, the fractional contributions to the average precipitation rate from ranges of intensity bins shown in the labels above each panel for (left column) timestep data and (right column) 3-hr means.

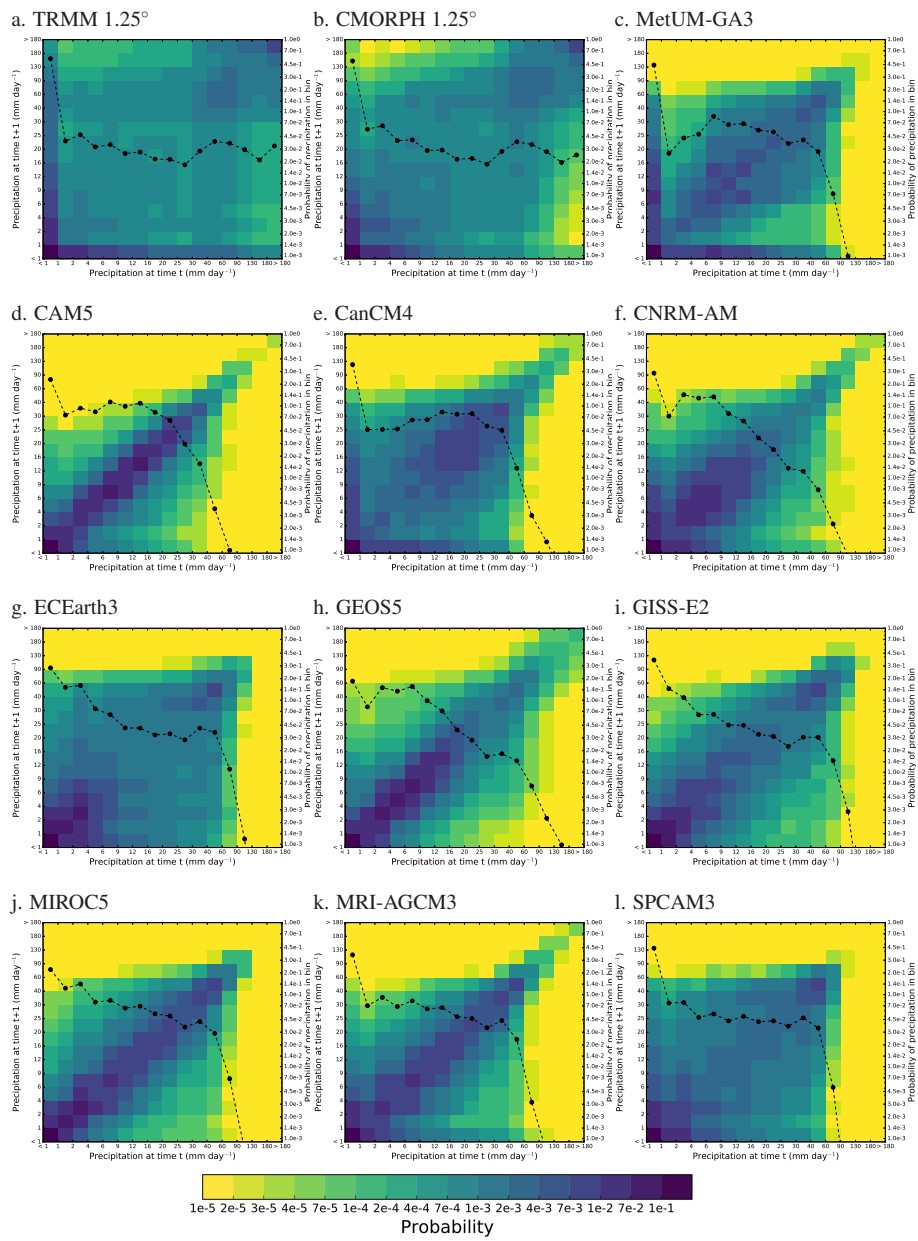

**Figure 10.** As in Fig. 4, but using 3-hr mean rain rates instead of timestep rain rates and with (a) TRMM 3B42 and (b) CMORPH 3-hr rain rates for the same temporal period and horizontal domain as the models.

a. Spatial — timestep, native grid

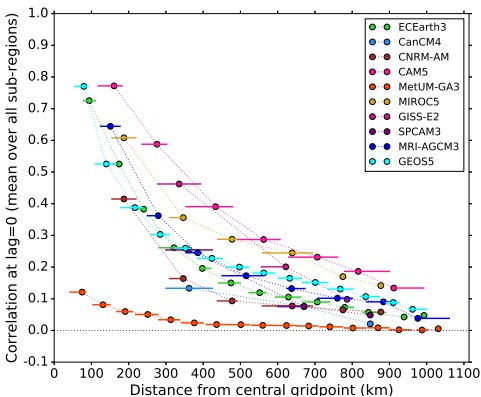

b. Spatial — 3-hr means, native grid

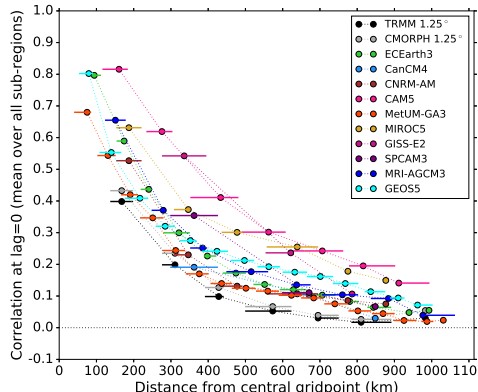

**Figure 11.** As in Fig. 6a, but for gridpoint precipitation data for (a) the native timestep and (b) 3-hr means. Panel (b) includes the TRMM 3B42 and CMORPH analyses at 1.25° resolution, for the same temporal period and horizontal domain as the models. Panel (a) is repeated from Fig. 6a for ease of comparison.

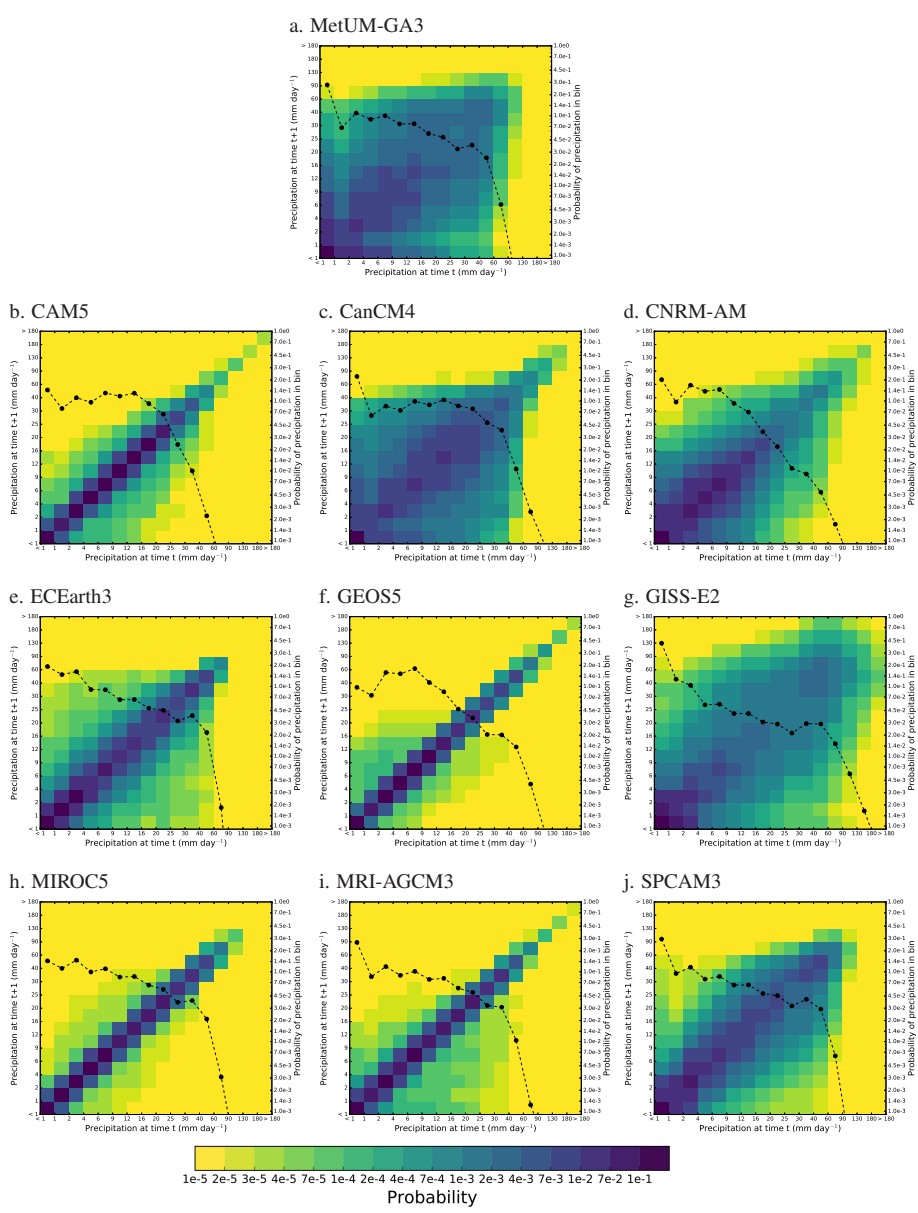

**Figure 12.** As in Fig. 4, but using timestep rain rates that were first spatially averaged to a $5.6° \times 5.6°$ horizontal grid.

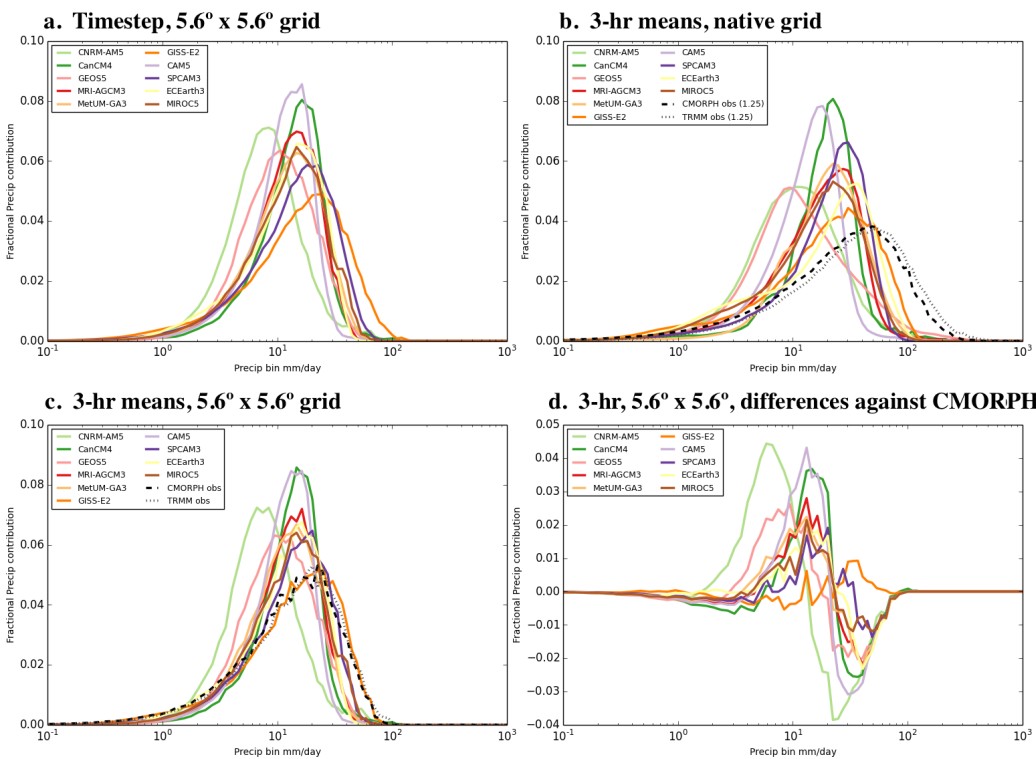

**Figure 13.** As in Fig. 8, but for (a) timestep precipitation rates averaged to a $5.6° \times 5.6°$ grid, and (b–d): 3-hr precipitation rates on (b) the native horizontal grid and (c,d) averaged to a $5.6° \times 5.6°$ grid. Panel (d) shows differences for each model minus CMORPH, using the $5.6° \times 5.6°$ data. Panel (b) is repeated from Fig. 8 for ease of comparison.

a. Temporal — timestep, native grid

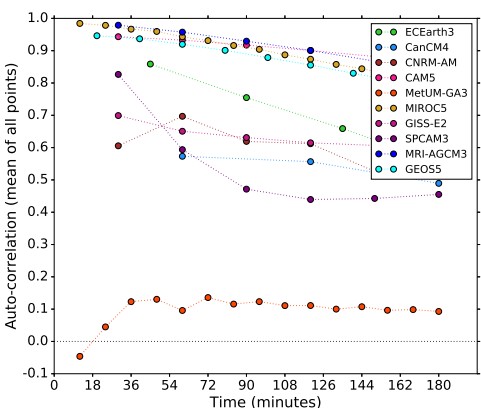

b. Temporal — timestep, 5.6°×5.6° grid

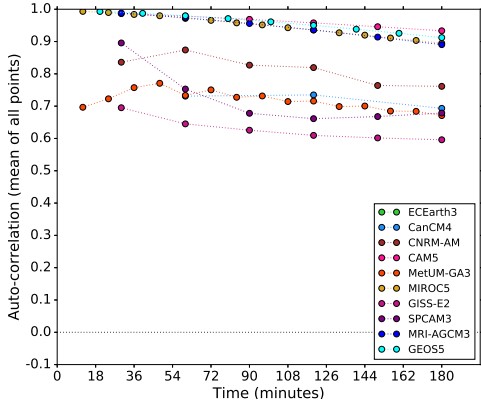

**Figure 14.** As in Fig. 6b, but for timestep rain rates at (a) the native gridscale and (b) averaged to a 5.6°×5.6° grid. Panel (a) is repeated from Fig. 6b for ease of comparison.

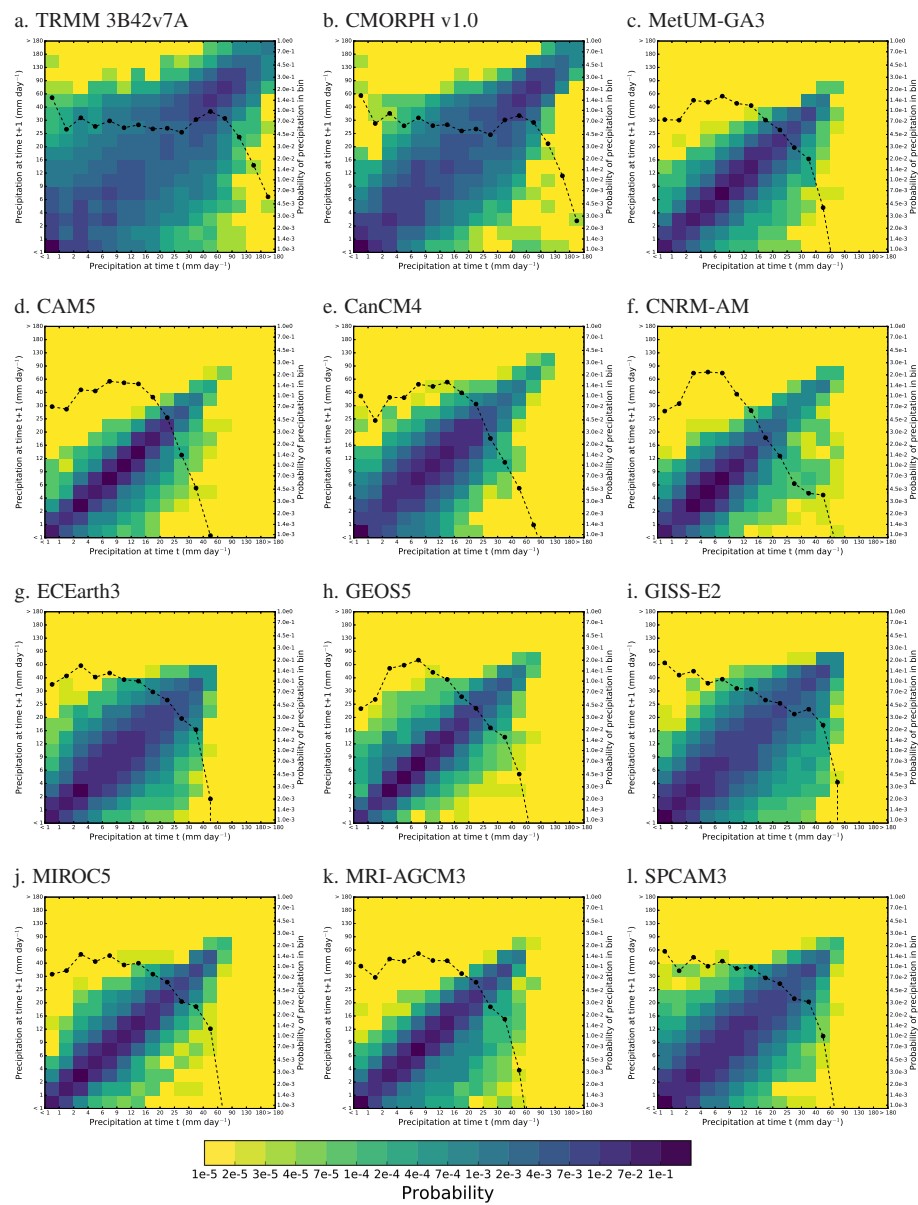

**Figure 15.** As in Fig. 10, but using 3-hr mean rain rates spatially averaged to a $5.6° \times 5.6°$ horizontal grid.

**Table 2.** For each model, as well as TRMM 3B42 and CMORPH: the number of timesteps in three hours; the dimensions (in native gridpoints) of the $5.6°\times5.6°\times$ averaging regions discussed in the text, with the total number of native gridpoints averaged together shown in parentheses; the number of $7\times7$ native-gridpoint regions in the analysis domain; and the number of $\approx1500$ km$\times1500$ km regions in the analysis domain, with the dimensions (in native gridpoints) on each side of the region shown in parentheses. Note that while GISS-E2 and SPCAM3 have the same resolution, they have different numbers of $7\times7$ gridpoint and $1500\times1500$ km regions because of the staggering of their native grids relative to the $10°$S–$10°$N, $60°$E–$160°$ analysis region.

| Model | $\Delta$t in 3hr | Lon×lat (total) in $5.6°\times5.6°$ | # 7×7 regions | # 1500 km (# points: lon × lat) |
|---|---|---|---|---|
| CAM5 | 6 | 4×6 (24) | 33 | 7 (10×12) |
| CanCM4 | 3 | 2×2 (4) | 5 | 6 (5×5) |
| CNRM-AM | 6 | 4×4 (16) | 20 | 7 (9×9) |
| ECEarth3 | 4 | 8×8 (64) | 80 | 7 (19×19) |
| GEOS5 | 9 | 9×11 (99) | 110 | 7 (21×27) |
| GISS-E2 | 6 | 2×2 (4) | 6 | 9 (5×5) |
| MetUM-GA3 | 15 | 10×15 (150) | 175 | 7 (24×35) |
| MIROC5 | 6 | 4×4 (16) | 20 | 7 (9×9) |
| MRI-AGCM3 | 6 | 5×5 (25) | 24 | 7 (12×12) |
| SPCAM3 | 6 | 2×2 (4) | 5 | 7 (5×5) |
| CMORPH 0.25° | 1 | 22×22 (484) | 748 | 8 (55×55) |
| CMORPH 1.25° | 1 | 4×4 (16) | 22 | 8 (10×10) |
| TRMM 3B42 0.25° | 1 | 22×22 (484) | 748 | 8 (55×55) |
| TRMM 3B42 1.25° | 1 | 4×4 (16) | 22 | 8 (10×10) |

**Table 3.** For each model, as well as TRMM 3B42 and CMORPH: summary metrics of spatial and temporal coherence in precipitation, using timestep and 3-hr data on the native horizontal grid and interpolated to a common $5.6° \times 5.6°$ horizontal grid. Positive values indicate that coherence is more common than intermittency; negative values indicate that intermittency is more common than coherence. Higher magnitudes indicate stronger coherence or intermittency for positive or negative values, respectively. The timestep column is marked "N/A" for TRMM and CMORPH because these datasets exist only as 3-hr values. By definition, the $5.6° \times 5.6°$ values are identical for the TRMM 0.25° and 1.25° datasets, as well as for the CMORPH 0.25° and 1.25° datasets.

| Model | Spatial coherence | | | | Temporal coherence | | | |
|---|---|---|---|---|---|---|---|---|
| | Native grid | | $5.6° \times 5.6°$ grid | | Native grid | | $5.6° \times 5.6°$ grid | |
| | Timestep | 3-hr | Timestep | 3-hr | Timestep | 3-hr | Timestep | 3-hr |
| CAM5 | 0.77 | 0.80 | 0.59 | 0.43 | 0.88 | 0.76 | 0.93 | 0.82 |
| CanCM4 | 0.23 | 0.28 | 0.47 | 0.30 | 0.38 | 0.57 | 0.46 | 0.66 |
| CNRM-AM | 0.41 | 0.52 | 0.41 | 0.35 | 0.44 | 0.53 | 0.59 | 0.71 |
| ECEarth3 | 0.67 | 0.73 | 0.51 | 0.38 | 0.72 | 0.57 | 0.83 | 0.68 |
| GEOS5 | 0.75 | 0.83 | 0.54 | 0.42 | 0.77 | 0.70 | 0.93 | 0.81 |
| GISS-E2 | 0.54 | 0.55 | 0.57 | 0.45 | 0.69 | 0.68 | 0.68 | 0.69 |
| MetUM-GA3 | -0.06 | 0.76 | 0.42 | 0.48 | 0.21 | 0.55 | 0.49 | 0.79 |
| MIROC5 | 0.65 | 0.67 | 0.61 | 0.48 | 0.92 | 0.71 | 0.95 | 0.81 |
| MRI-AGCM3 | 0.65 | 0.66 | 0.51 | 0.39 | 0.91 | 0.69 | 0.93 | 0.78 |
| SPCAM3 | 0.33 | 0.43 | 0.55 | 0.33 | 0.71 | 0.56 | 0.74 | 0.68 |
| CMORPH 0.25° | N/A | 0.80 | N/A | 0.34 | N/A | 0.41 | N/A | 0.73 |
| CMORPH 1.25° | N/A | 0.55 | N/A | 0.34 | N/A | 0.50 | N/A | 0.73 |
| TRMM 3B42 0.25° | N/A | 0.69 | N/A | 0.32 | N/A | 0.29 | N/A | 0.68 |
| TRMM 3B42 1.25° | N/A | 0.49 | N/A | 0.32 | N/A | 0.39 | N/A | 0.68 |