# Peer review of "ASoP (v1.0): A set of methods for analyzing scales of precipitation in general circulation models"

_Geoscientific Model Development, 2016_

## Referee Comment (RC1) · C. Jakob (Referee) · 15 Sep 2016

C. Jakob (Referee)

christian.jakob@monash.edu

General Comments

This is an excellent paper that introduces a number of interesting and useful diagnostics to study the behaviour of precipitation in weather and climate models at various space and time-scales all the way to single grid points and time steps. The paper is very well written and the analyses and arguments are sound. It adds several new ideas for model analysis and evaluation, all of which will add to the arsenal available to the community. I have a few minor comments, which I list below.

Specific Comments

[Figure]

1) The new diagnostics introduce are very nice and worth looking at in the present form. However, as there are many of them, it would be nice if the authors could consider producing a few summary measures that could be presented more easily when comparing many models and/or evaluating them against observations. This would contribute to the growing interest in having "performance metrics", so that changes in models can be more easily assessed and their effects quantified. I am aware that there is no single metric that identifies good or bad models, but by having a collection of them - well beyond this study - I believe the community will be able to better communicate model improvement in the future. For one-dimensional histograms, there are simple statistical techniques one could use, such as the Kolmogorov-Smirnov two sample test, which allows an assessment of the likelihood that two samples are drawn from the same population. This would be especially useful where models are compared to observations. It would be nice to also have a summary measure of the two-dimensional histograms presented here, but that might be more difficult.

2) Page 12, Line 6-7: This is a very strange argument. It's likely you did look at the lagged time correlations for 3-hourly rainfall and found something. Why not just state what you found - no need to show a figure if you can say it in words. Saying that there could be a problem, rather than that there is one, sounds like you have something to hide.

3) Page 13, Line 28-29: The atmosphere is not in radiative convective equilibrium at the scale of 600 km over 3 hours. If the models were, that would be completely wrong. To provide evidence, I attach an unpublished figure from my own work, which plots daily averages of atmospheric cooling derived from CERES observations against daily averages of rainfall from GPCP for increasing areas centered on a 1x1 degree gridpoint in the Tropical Western Pacific. If we consider the 90x90 degree area as close to RCE, it is evident that the averaging scale one could argue starts to approach it is about 30-40 degrees (i.e., (3000-4000 km)^2), far from the 600 km scale speculated about here. In fact, not surprisingly, at those scales, the atmosphere is a far from

[Figure]

RCE as it can be, as the heavy rainfall is associated with large cloud fields that reduce the atmospheric radiative cooling substantially. More likely, at the 600 km scale the convection is in balance with dynamical systems at the synoptic scale, which do exist in the tropics, and are perhaps relatively well captured by the models.

Technical Comments

4) Page 6, Line 7-8 and Figures 2 and 5: The bin below 0.5 * Delta x appears pointless and makes for an awkward plot. I suggest changing the txt to acknowledge the theoretical possibility of the existence of such a bin but then states that in practice it does not exist for this study. Then you can remove the unnecessary and distracting XXX columns from the figures.

—————————————————————

Fig. 1.

[Figure]

---

## Referee Comment (RC2) · Anonymous Referee #2 · 13 Oct 2016

General comments: This paper proposes a diagnosis method for precipitation of general circulation models (GCMs) by using a native temporal and spatial grids and discusses dependency of temporal and spatial averages of precipitation. Precipitation behaviors of GCMs have been usually evaluated by climatological mean states. However, this study clearly shows that even if the climatological mean (or 3-hr average) precipitation is almost the same, its temporal and spatial behaviors are very different if analyzed by the native grids and original time step. This aspect of precipitation might affect large scale behaviors and hence must be more focused for the analysis, evaluations, or improvements of GCMs. The methodology is clear, and it implication is sound. Thus, I suggest publication of this study after minor revisions described below. Although the

proposed diagnosis will be useful, the authors can go further more. In the text, the authors mention "persistency" or "intermittency" of precipitation. We need to compare many figures (e.g. Fig. 4 vs Fig. 7) to evaluate "persistency" or "intermittency". The authors should consider some quantifications of "persistency" and "intermittency", and show summary of these quantities of the models with difference samplings.

Specific comments

p. 3, L18, "Such diagnostics": It is not clear which "diagnostics" is referred to in this paragraph. Please clarify.

p. 4, L5-6, "Both products are derived from a combination of infrared and microwave sounders and calibrated against gauge data.": The authors should add more information on the difference between TRMM and CMORPH for readers who are not familiar to the details of the products of precipitation. In addition, since TRMM 3B42 is not solely based on the TRMM data, it is not appropriate to call it "TRMM". The authors should make a remark on it if the abbreviation of "TRMM" is to be used.

p. 5, L23, "We find the central point in each region and extract the timeseries of precipitation.": I suggest that "find" should be replaced by "define" or an appropriate word.

p. 5, L26, "in Figs. 2b and 2c for CMORPH": These should be "Figs. 2c and 2d".

p. 6, L10, "these computations result in a matrix of correlations with distance and time, as shown in Fig. 2c.": I guess that this is for Figs. 2e and 2f.

p. 6, L14, "Fig. 2b": This should be replaced by "Figs. 2c and 2d".

p. 6, L16, "For the ranges shown here, the CMORPH 0.25_ correlations decline more quickly with time than with space.": It is ambiguous to say which is "more quick" between time and space. Add more explanations.

p. 8, L20, "The 1D histogram suggests that MetUM-GA3 oscillates between lighter ($< 9$ mm day$-1$) and heavier ($> 30$ mm day$-1$) rain rates, with almost no instances

of moderate rates (9–30 mm day−1).": This is an interesting behavior of precipitation of the MetUM-GA3. Please consider adding sample figures of time sequence and snapshot distribution of precipitation of MetUM-GA3.

p. 8, L25, "The bi-modal 1D histogram suggests that most deep convection in MetUM-GA3 is strong.": It is not clear how "strong" or stronger than what? Please add more explanations.

p. 9, L10: "Despite having the finest horizontal resolution" should be replaced by "Because having the finest horizontal resolution"?

p. 9, L13: After "The lag-1 correlation at the central gridpoint is slightly negative", add "for MetUM-GA3" for readability.

p. 9, L19: Delta is superscript. Please correct such that "0.5–1.5$\Delta$x".

p. 10, L33, "While there were no observation-based constraints on timestep rainfall": This statement is incorrect. We can use the ground radar data for very high-spatial and temporal resolution of precipitation, such as 1 km and 10 min. We can also use satellite radar data for high-spatial distribution of precipitation, such as PR of TRMM or DPR of GPM. The authors should add discussions on using and analyzing such high-resolution radar data for evaluations of precipitation in future directions.

p. 10, L34, "Both TRMM and CMORPH produce histograms that are broader than the models' histograms and which peak at heavier precipitation rates.": These observation also have biases especially for lighter rainfalls. The authors should add remark on the biases of the observations in the earlier sections such as in the methodology.

p. 11, L8, "(dashed line on Fig.s 4a)": "Fig.s 4a should be "Fig. 4a".

p. 11, L15, "Conversely, models with more persistent timestep precipitation (e.g., GEOS5, MRI-AGCM, CAM5 and MIROC5) display greater intermittency for 3-hr means.": To clarify the sentence, please add "when Fig. 9 is compared with Fig. 4". It is not clear how great the intermittency is. The authors should quantify the intermittency.

p. 11, L28-29, "SPCAM3, ECEarth3 and CanCM4 are perhaps closest to TRMM and CMORPH, but are still more persistent.": Again, it is not clear how these models are close to the observations. Please consider quantification of the persistency.

p. 11, L34, "We note that there are also differences between TRMM and CMORPH over this short period: CMORPH displays more frequent light precipitation than TRMM, which has been shown to under-detect light rainfall (Huffman et al., 2007, e.g.,). TRMM is more intermittent than CMORPH.": The authors should note why these differences come from between the two observations. "(Huffman et al., 2007, e.g.,)" should be "(e.g., Huffman et al., 2007)".

p. 12, L3, "All models display higher correlations": Add "at 3h interval (Fig. 10b)" at the end of this sentence.

p. 12, L11, "Spatial averaging reduces timestep intermittency in all models (Fig. 11).": The observations of TRMM and CMORPH should also be added to Fig. 11.

p. 12, L18-20, "This suggests that using a common horizontal grid or a common timescale does not necessarily create a fair comparison between models, due to differences in the number of points or timesteps, respectively, that are combined to create the average.": It is not appropriate to say "a fair comparison", because it is not clear in what sense the fair comparison means. Use of a common horizontal grid or a common timescale has its importance for some purposes. Please rephrase this sentence.

p. 12, L24, "the comparison of Fig. 4a and Fig. 11a suggests that MetUM-GA3 likely has only a few precipitating gridpoints . . .": This sentence is not clear. Please explain what the authors want to mean.

p. 12, L28, "By contrast, the comparison of Fig. 4f and Fig. 11f": Add "(GEOS5)" for clarification.

p. 12, L39, "in Figs. 7a and 12a) implies": Delete ")".

p. 14, L9, "Although there are no verifying observations for our timestep data": As

mentioned before (p. 4, L5-6), the ground radar data can be used for verification of the timestep data.

p. 14, L33-34: "Fig. 4a" and "Fig. 4b" should be "Fig. 7a" and "Fig. 7b", respectively.

p. 14, Section 4 Discussion: For understanding of properties of cumulus convection schemes, single column models (SCM) have been widely used. Especially, SCM under a radiative-convective equilibrium (RCE) condition is a useful framework for understanding the timestep behaviors. For example, Satoh and Hayashi (1992, J. Atmos. Sci.), Takata and Noda (1997, J. Meteor. Soc. Japan) for SCM in RCE. Please add discussions on the above aspects of using a SCM for understanding of intermittency.

-Satoh, M., and Hayashi, Y. Y. (1992). Simple cumulus models in one-dimensional radiative convective equilibrium problems. Journal of the Atmospheric Sciences, 49, 1202-1220. -Takata, K., and Noda, A. (1997). The effect of cumulus convection on $CO_2$-induced climate change in the tropics. Journal of the Meteorological Society of Japan, 75, 677-686.

---

## Author Comment (AC1) · 23 Nov 2016

**GMD-2016-161**
Response to comments from Christian Jakob

We thank the reviewer for providing a positive and encouraging review of our manuscript. In our response below, the reviewer's comments appear in red text; our response to the reviewer's comments appears in black text.

*General comments*

This is an excellent paper that introduces a number of interesting and useful diagnos-

tics to study the behaviour of precipitation in weather and climate models at various space and time-scales all the way to single grid points and time steps. The paper is very well written and the analyses and arguments are sound. It adds several new ideas for model analysis and evaluation, all of which will add to the arsenal available to the community. I have a few minor comments, which I list below.

We thank the reviewer for these positive comments.

*Specific comments*

1) The new diagnostics introduce are very nice and worth looking at in the present form. However, as there are many of them, it would be nice if the authors could consider producing a few summary measures that could be presented more easily when comparing many models and/or evaluating them against observations. This would contribute to the growing interest in having "performance metrics", so that changes in models can be more easily assessed and their effects quantified. I am aware that there is no single metric that identifies good or bad models, but by having a collection of them—well beyond this study—I believe the community will be able to better communicate model improvement in the future. For one-dimensional histograms, there are simple statistical techniques one could use, such as the Kolmogorov-Smirnov two sample test, which allows an assessment of the likelihood that two samples are drawn from the same population. This would be especially useful where models are compared to observations. It would be nice to also have a summary measure of the two-dimensional histograms presented here, but that might be more difficult.

We agree with the reviewer and thank the reviewer for this suggestion. Reviewer #2 also proposed that we quantify the persistence and intermittency of precipitation in models and observations. We have combined the two reviewers' suggestions and provide a unified response below, a copy of which appears in our response to Reviewer #2.

We have created summary metrics of spatial and temporal coherence in precipitation, which allow the reader to more easily evaluate models, either against each other or against observations. These metrics are based on the persistence of upper- and lower-quartile precipitation in time (measured from one timestep to the next) or space (measured at neighboring gridpoints). Considered together, the metrics summarise aspects of our two-dimensional histograms, as well as our correlations of precipitation as functions of distance and time, but without relying on the choice of a threshold correlation value or spatial or temporal scale. The metrics are scaled to range from -1 to +1: positive values indicate that persistence is more common than intermittency; negative values indicate that intermittency is more common than persistence. Table 3 of the revised manuscript shows these metrics for all models, as well as for TRMM and CMORPH satellite-derived observations, and for all horizontal grids and temporal frequencies considered in our study (i.e., timestep and 3-hr, native-grid and $5.6° \times 5.6°$ averages). The metrics confirm many of the conclusions of our study concerning the effects of averaging in space and time on the spatial and temporal coherence of simulated precipitation features. These quantitative metrics also confirm our qualitative conclusions about the relative levels of spatial and temporal coherence in the models we analysed.

We have added a description of these metrics to Section 2.1.4. The metrics are shown in Table 3. We discuss the metrics throughout the Results section (Section 3) and refer to them in the Discussion (Section 4) and Conclusions (Section 5) sections.

These metrics have helped to demonstrate quantitatively the qualitative conclusions we had drawn from our analysis, so we are very grateful to the reviewers for suggesting them.

2) Page 12, Line 6-7: This is a very strange argument. It's likely you did look at the lagged time correlations for 3-hourly rainfall and found something. Why not just state what you found—no need to show a figure if you can say it in words. Saying that there

could be a problem, rather than that there is one, sounds like you have something to hide.

We agree with the reviewer. We have changed the paragraph in question to read: "For model data, lagged correlations of 3-hr precipitation (i.e., as in Fig. 6b but for 3-hr data) over a 36-hr window were dominated by the overly strong and regular diurnal cycle of precipitation in the models, which manifested itself in our diagnostics as a pronounced peak in the correlations at a 24-hr lag (not shown). TRMM and CMORPH displayed a much weaker and broader peak across lags of 18–30 hr, suggesting greater day-to-day variability in the timing of the diurnal maximum in tropical maximum in the satellite-observations than in the models."

We believed that because the lag-correlation diagrams for 3-hr data were dominated by the diurnal cycle in the models, showing the diagrams would not address our original objective of analysing the temporal coherence in the 3-hr precipitation data on sub-daily scales (i.e., lag-1 or lag-2 correlations in the 3-hr data). Instead, it would unduly focus the reader's attention on the errors in the diurnal cycle, which are outside the scope of our study.

3) Page 13, Line 28-29: The atmosphere is not in radiative convective equilibrium at the scale of 600 km over 3 hours. If the models were, that would be completely wrong. To provide evidence, I attach an unpublished figure from my own work, which plots daily averages of atmospheric cooling derived from CERES observations against daily averages of rainfall from GPCP for increasing areas centered on a 1x1 degree gridpoint in the Tropical Western Pacific. If we consider the 90x90 degree area as close to RCE, it is evident that the averaging scale one could argue starts to approach it is about 30-40 degrees (i.e., $(3000\text{-}4000 \text{ km})^2$), far from the 600 km scale speculated about here. In fact, not surprisingly, at those scales, the atmosphere is a far from RCE as it can be, as the heavy rainfall is associated with large cloud fields that reduce the atmospheric radiative cooling substantially. More likely, at the 600 km scale the convection is in

balance with dynamical systems at the synoptic scale, which do exist in the tropics, and are perhaps relatively well captured by the models.

We agree with the reviewer and thank the reviewer for this suggestion. We have replaced the sentence in question with: "We hypothesize that these broader scales represent those at which simulated convection is in balance with the synoptic-scale, dynamical systems that produce precipitation, predictions of which should be highly similar among the models in the short, 2-day hindcasts we analysed."

We replaced a similar sentence in the Discussion section with: "This convergence of model behavior may be enhanced by the fact that these data are from short (2-day) forecasts initialized from the same ECMWF analyses, which means that the representation of these dynamical systems are much more similar among models than if the data came from free-running climate simulations."

4) Page 6, Line 7-8 and Figures 2 and 5: The bin below 0.5 * Delta x appears pointless and makes for an awkward plot. I suggest changing the txt to acknowledge the theoretical possibility of the existence of such a bin but then states that in practice it does not exist for this study. Then you can remove the unnecessary and distracting XXX columns from the figures.

We agree with the reviewer. In the revised manuscript, we have removed the bin below $0.5\Delta x$ from our analysis and the column from our revised versions of Figs. 2 and 5.

---

## Author Comment (AC2) · 23 Nov 2016

**GMD-2016-161**
Response to comments from Anonymous Referee #2

We thank the reviewer for providing a thorough and generally positive review of our manuscript. In our response below, the reviewer's comments appear in red text; our response to the reviewer's comments appears in black text.

*General comments*

This paper proposes a diagnosis method for precipitation of general circulation models

(GCMs) by using a native temporal and spatial grids and discusses dependency of temporal and spatial averages of precipitation. Precipitation behaviors of GCMs have been usually evaluated by climatological mean states. However, this study clearly shows that even if the climatological mean (or 3-hr average) precipitation is almost the same, its temporal and spatial behaviors are very different if analyzed by the native grids and original time step. This aspect of precipitation might affect large scale behaviors and hence must be more focused for the analysis, evaluations, or improvements of GCMs. The methodology is clear, and it implication is sound. Thus, I suggest publication of this study after minor revisions described below. Although the proposed diagnosis will be useful, the authors can go further more. In the text, the authors mention "persistency" or "intermittency" of precipitation. We need to compare many figures (e.g. Fig. 4 vs Fig. 7) to evaluate "persistency" or "intermittency". The authors should consider some quantifications of "persistency" and "intermittency", and show summary of these quantities of the models with difference samplings.

We agree with the reviewer and thank the reviewer for this suggestion. Reviewer #1 (Christian Jakob) also proposed that we quantify the persistence and intermittency of precipitation in models and observations. We have combined the two reviewers' suggestions and provide a unified response below, a copy of which appears in our response to Reviewer #1.

We have created summary metrics of spatial and temporal coherence in precipitation, which allow the reader to more easily evaluate models, either against each other or against observations. These metrics are based on the persistence of upper- and lower-quartile precipitation in time (measured from one timestep to the next) or space (measured at neighboring gridpoints). Considered together, the metrics summarise aspects of our two-dimensional histograms, as well as our correlations of precipitation as functions of distance and time, but without relying on the choice of a threshold correlation value or spatial or temporal scale. The metrics are scaled to range from -1 to +1: positive values indicate that persistence is more common than intermittency;
negative values indicate that intermittency is more common than persistence. Table 3 of the revised manuscript shows these metrics for all models, as well as for TRMM and CMORPH satellite-derived observations, and for all horizontal grids and temporal frequencies considered in our study (i.e., timestep and 3-hr, native-grid and $5.6° \times 5.6°$ averages). The metrics confirm many of the conclusions of our study concerning the effects of averaging in space and time on the spatial and temporal coherence of simulated precipitation features. These quantitative metrics also confirm our qualitative conclusions about the relative levels of spatial and temporal coherence in the models we analysed.

We have added a description of these metrics to Section 2.1.4. The metrics are shown in Table 3. We discuss the metrics throughout the Results section (Section 3) and refer to them in the Discussion (Section 4) and Conclusions (Section 5) sections.

These metrics have helped to demonstrate quantitatively the qualitative conclusions we had drawn from our analysis, so we are very grateful to the reviewers for suggesting them.

*Specific comments*

p. 3, L18, "Such diagnostics": It is not clear which "diagnostics" is referred to in this paragraph. Please clarify.

We were referring to diagnostics that estimate the coherence of precipitation in space and time, which we mentioned in the previous sentence. In the revised version of the manuscript, we have replaced "such diagnostics" with "diagnostics of precipitation coherence".

p. 4, L5-6, "Both products are derived from a combination of infrared and microwave sounders and calibrated against gauge data.": The authors should add more information on the difference between TRMM and CMORPH for readers who are not familiar to the details of the products of precipitation. In addition, since TRMM 3B42 is not solely

based on the TRMM data, it is not appropriate to call it "TRMM". The authors should make a remark on it if the abbreviation of "TRMM" is to be used.

We disagree with the reviewer that it is inappropriate to call the 3B42 product "TRMM". In fact, the creators of the dataset, the U.S. National Aeronautics and Space Administration (NASA), refer to the product as "TRMM 3B42". Please see their webpage here:

http://disc.sci.gsfc.nasa.gov/precipitation/documentation/TRMM_README/TRMM_3B42_readme.shtml

The Tropical Rainfall Measuring Mission (TRMM) is a project, which includes a satellite that is—confusingly, we agree—also called TRMM. The microwave instrument on the satellite is often called the TRMM Microwave Imager (TMI). Some of the products that TRMM (the project) produces use data exclusively from TRMM (the satellite), while other TRMM (the project) products use data from multiple instruments, including TRMM (the satellite). The 3B42 product falls into the latter category. Many studies, and NASA itself, refer to the 3B42 product as "TRMM 3B42." It is not our place to correct this confusion; rather, we use the commonly accepted nomenclature.

We have added details to Section 2 about the TRMM 3B42 and CMORPH algorithms, including differences in how they produce their merged microwave–infrared precipitation estimates. We have also added a note that states that the TRMM 3B42 product is produced from multiple microwave sounders, not solely TMI. We have also added "3B42" to all figure captions and tables where we use TRMM 3B42 data.

p. 5, L23, "We find the central point in each region and extract the timeseries of precipitation.": I suggest that "find" should be replaced by "define" or an appropriate word.

We have replaced "find" with "select". We hope that this is acceptable.

p. 5, L26, "in Figs. 2b and 2c for CMORPH": These should be "Figs. 2c and 2d".

We agree with the reviewer. We have corrected this error by replacing "in Figs. 2b

and 2c" with "in Figs. 2c and 2d".

p. 6, L10, "these computations result in a matrix of correlations with distance and time, as shown in Fig. 2c.": I guess that this is for Figs. 2e and 2f.

We agree with the reviewer. We have corrected this error by replacing "in Fig. 2c" with "in Figs. 2e and 2f".

p. 6, L14, "Fig. 2b": This should be replaced by "Figs. 2c and 2d".

We agree with the reviewer. We have corrected this error by replacing "in Fig. 2b" with "in Figs. 2c and 2d".

p. 6, L16, "For the ranges shown here, the CMORPH 0.25° correlations decline more quickly with time than with space.": It is ambiguous to say which is "more quick" between time and space. Add more explanations.

We thank the reviewer for pointing out this ambiguity. On reflection, this sentence does not contribute anything meaningful to our discussion and is a likely source of confusion. We have removed it from the revised manuscript.

p. 8, L20, "The 1D histogram suggests that MetUM-GA3 oscillates between lighter ($< 9$ mm day$^{-1}$) and heavier ($> 30$ mm day$^{-1}$) rain rates, with almost no instances of moderate rates (9–30 mm day$^{-1}$).": This is an interesting behavior of precipitation of the MetUM-GA3. Please consider adding sample figures of time sequence and snapshot distribution of precipitation of MetUM-GA3.

We agree with the reviewer. We have added a figure to our revised manuscript (Fig. 7) that compares MetUM-GA3 to GEOS5. GEOS5 produces persistent rainfall and has a timestep and horizontal resolution similar to MetUM-GA3. In Fig. 7, we compare timeseries of precipitation at an example gridpoint in the Indian Ocean (0°, 90°E) for an example 48-hour forecast from our datasets (4 November 2010); we also compare snapshots of instantaneous precipitation rates for an example timestep (20:00 UTC

on 4 November 2010). These figures confirm the results of our other diagnostics: precipitation in MetUM-GA3 is temporally and spatially intermittent, while precipitation in GEOS5 is temporally and spatially persistent. We have added a paragraph to Section 3.1 of our revised manuscript that discusses this figure and its implications.

We thank the reviewer for this suggestion, which has helped to provide further visual evidence of our main conclusions.

p. 8, L25, "The bi-modal 1D histogram suggests that most deep convection in MetUM-GA3 is strong.": It is not clear how "strong" or stronger than what? Please add more explanations.

We agree with the reviewer that this statement is not clear. We mean that most deep convection in MetUM-GA3 is as intense as it possibly can be, at that horizontal resolution and scientific configuration of the model. We have modified this sentence to state that "most deep convection in MetUM-GA3 is as strong as possible, given the horizontal resolution and scientific configuration of the model."

p. 9, L10: "Despite having the finest horizontal resolution" should be replaced by "Because having the finest horizontal resolution"?

We disagree with the reviewer's suggestion. As our explanation in the sentence following the one that the reviewer quoted states, we would expect (naïvely, perhaps) that finer horizontal resolution would increase spatial correlations, particularly when those correlations are measured as a function of the native gridpoint. A finer-resolution model has a smaller physical distance between gridpoints, which means that a precipitation feature of the same physical dimension would span more gridpoints in a finer-resolution model than in a coarser-resolution model. Thus, one would expect the finer-resolution model to have a higher spatial correlation, measured in native gridpoints. MetUM-GA3 has the lowest spatial correlation and the finest spatial resolution, which is at odds with this expectation. Therefore, we have maintained our use of "despite".

p. 9, L13: After "The lag-1 correlation at the central gridpoint is slightly negative", add "for MetUM-GA3" for readability.

We agree with the reviewer. We have added "for MetUM-GA3" to the end of this sentence.

p. 9, L19: Delta is superscript. Please correct such that "0.5–1.5△x".

We agree with the reviewer. We have corrected the formatting of the △ symbol.

p. 10, L33, "While there were no observation-based constraints on timestep rainfall": This statement is incorrect. We can use the ground radar data for very high-spatial and temporal resolution of precipitation, such as 1 km and 10 min. We can also use satellite radar data for high-spatial distribution of precipitation, such as PR of TRMM or DPR of GPM. The authors should add discussions on using and analyzing such high-resolution radar data for evaluations of precipitation in future directions.

We agree with the reviewer. We were referring specifically to the lack of verifying observations for similar spatial domains and time periods to those used in our analysis of the 2-day hindcast data from the MJO inter-comparison project. We acknowledge that there are high-resolution observations, such as those the reviewer cites, that could be compared against model simulations, assuming care was taken to perform those comparisons at similar spatial and temporal resolutions, based on the results from our study. In fact, the comparison of high-resolution, convection-permitting model simulations against radar data is an area of ongoing research.

We have modified the sentence in question to read: "While there are no observation-based constraints on timestep rainfall for similar spatial domains and temporal periods as the model data analysed here . . . ". We believe this statement is accurate.

Further, we have added a short paragraph to the Discussion section that discusses the use of high-resolution radar data to validate simulated timestep precipitation, as the

reviewer suggests. This is the second paragraph of the Discussion section (Section 4) in our revised manuscript.

p. 10, L34, "Both TRMM and CMORPH produce histograms that are broader than the models? histograms and which peak at heavier precipitation rates.": These observation also have biases especially for lighter rainfalls. The authors should add remark on the biases of the observations in the earlier sections such as in the methodology.

We agree with the reviewer. We have added a sentence to Section 2 to note this bias: "Both products have been shown to under-detect light rainfall rates (e.g., Tian et al., 2010)." Further, we note that we commented on the under-detection of light rainfall on page 11, line 34 of the original manuscript. We have added the Tian et al. (2010) reference above to that sentence as well.

We agree with the reviewer. We have removed the erroneous "s" after "Fig.".

p. 11, L15, "Conversely, models with more persistent timestep precipitation (e.g., GEOS5, MRI-AGCM, CAM5 and MIROC5) display greater intermittency for 3-hr means.": To clarify the sentence, please add "when Fig. 9 is compared with Fig. 4". It is not clear how great the intermittency is. The authors should quantify the intermittency.

We agree with the reviewer's first point. We have added the statement "(compare Fig. 9 to Fig. 4)" to the end of the sentence in question.

We also agree with the reviewer's second point. We have added metrics for spatial co-herence and temporal persistence to our revised manuscript (see response to "General Comments" above). These metrics clearly demonstrate that averaging from timestep to 3-hr means increases the intermittency in models with more-persistent timestep pre-cipitation (see Table 3 in revised manuscript). We have added the following sentence to the end of the paragraph in question: "Table 3 confirms that temporal averaging on the

native grid reduces inter-model variations in the temporal persistence summary metric, by increasing values for models with relatively low scores (e.g., MetUM-GA3, CanCM4, CNRM-AM) and reducing the values for models with relatively high scores (e.g., CAM5, MIROC5, MRI-AGCM3)."

Please note that we revised the sentence in question to read "Conversely, models with more persistent timstep precipitation (e.g., GEOS5, MRI-AGCM, CAM5 and MIROC5) display reduced persistence when data are averaged to 3-hr means" to prevent the mis-understanding that these models were the most intermittent at the 3-hr among all the models considered. In fact, these models are still the most persistent models, as our new summary metrics show, but they show reduced persistence (and lower values of our metric) for 3-hr data than for timestep data.

p. 11, L28-29, "SPCAM3, ECEarth3 and CanCM4 are perhaps closest to TRMM and CMORPH, but are still more persistent.": Again, it is not clear how these models are close to the observations. Please consider quantification of the persistency.

We agree with the reviewer. As above, we have created new metrics to quantify temporal persistence and spatial coherence, which can be found in Table 3 of the revised manuscript. These metrics demonstrate that SPCAM3, ECEarth3, CanCM4 and CNRM-AM show temporal persistence in 3-hr precipitation that is most similar to TRMM and CMORPH, but that all models are too persistent with respect to both datasets. We have added references to our summary metrics to the paragraph in question, noting that these four models show values closest to the satellite-derived observations.

p. 11, L34, "We note that there are also differences between TRMM and CMORPH over this short period: CMORPH displays more frequent light precipitation than TRMM, which has been shown to under-detect light rainfall (Huffman et al., 2007, e.g.,). TRMM is more intermittent than CMORPH.": The authors should note why these differences come from between the two observations. "(Huffman et al., 2007, e.g.,)" should be "(e.g., Huffman et al., 2007)".

We disagree with the reviewer's first point. The purpose of our manuscript is not to understand the difference between the TRMM and CMORPH datasets. Because we do not have direct access to the algorithms used to produce the TRMM 3B42 and CMORPH products, we cannot investigate the reasons for the differences in temporal coherence between the two datasets. Any hypotheses would be mere conjecture that would not be worthy of publication. Instead, we compare these datasets only to provide a measure of observational uncertainty in our diagnostics. We have added a sentence to Section 2 to clarify this: "We employ two observation-based datasets to provide a measure of observational uncertainty in our diagnostics."

We agree with the reviewer's second point. We have corrected this errors by moving the "e.g.," to the beginning of the parenthetical citation.

p. 12, L3, "All models display higher correlations": Add "at 3h interval (Fig. 10b)" at the end of this sentence.

We agree with the reviewer. We have added "When using 3-hr data" to the start of the sentence in question and a reference to Fig. 10b at the end of the sentence.

p. 12, L11, "Spatial averaging reduces timestep intermittency in all models (Fig. 11).": The observations of TRMM and CMORPH should also be added to Fig. 11.

We disagree with the reviewer. It is not possible to show data for TRMM and CMORPH in Fig. 11, because Fig. 11 shows timestep data from models, not 3-hr data. Comparing timestep data (12–60 minutes) from models to 3-hr data from TRMM and CMORPH is not valid. Fig. 14 in our manuscript compares 3-hr data from TRMM and CMORPH to 3-hr data from models, when all datasets have also been interpolated to a $5.6° \times 5.6°$ horizontal grid. This is the equivalent of Fig. 11 for 3-hr data; it provides a fair comparison between models and satellite-derived observations.

p. 12, L18-20, "This suggests that using a common horizontal grid or a common timescale does not necessarily create a fair comparison between models, due to differ-

ences in the number of points or timesteps, respectively, that are combined to create the average.": It is not appropriate to say "a fair comparison", because it is not clear in what sense the fair comparison means. Use of a common horizontal grid or a common timescale has its importance for some purposes. Please rephrase this sentence.

We agree with the reviewer. We were referring specifically to the diagnostics we produced in this study. Of course, interpolating to a common grid or timescale is appropriate for some purposes, outside the remit of this study. We have amended this sentence to read "For the purposes of these diagnostics, using a common horizontal grid or a common timescale does not necessarily create a fair comparison between models . . . ".

p. 12, L24, "the comparison of Fig. 4a and Fig. 11a suggests that MetUM-GA3 likely has only a few precipitating gridpoints . . . ": This sentence is not clear. Please explain what the authors want to mean.

We do not know exactly which aspect of this sentence the reviewer finds unclear. Our point is that when one averages over a broad region, such as our $5.6° \times 5.6°$ boxes, two models can produce the same precipitation rate (or spectrum of rates) from different combinations of precipitation frequency and intensity on the native gridscale. For example, Model A might produce very infrequent, but very heavy precipitation at its native resolution. Model B might produce very frequent, but light precipitation. When data from these models are averaged across a $5.6° \times 5.6°$ box, the average precipitation rates might be the same. In our study, MetUM-GA3 is like Model A. Within each $5.6° \times 5.6°$ box at any given time, there are very few precipitating gridpoints, but those gridpoints show heavy precipitation. Our new Fig. 7—which the reviewer suggested—demonstrates this well.

We have revised this sentence to read "For instance, the comparison of Fig. 4a and Fig. 12a suggests that MetUM-GA3 likely has only a few precipitating gridpoints in each $5.6° \times 5.6°$ region, but that those points show very heavy precipitation (e.g., 90–130 mm day$^{-1}$), as indicated in Fig. 8a."

p. 12, L39, "in Figs. 7a and 12a) implies": Delete ")".

We agree with the reviewer. We have corrected this error by deleting the extraneous right-hand parenthesis.

p. 14, L9, "Although there are no verifying observations for our timestep data": As mentioned before (p. 4, L5-6), the ground radar data can be used for verification of the timestep data.

We agree with the reviewer. Again, we were referring to the lack of verifying observations for similar spatial domains and time periods as the ones covered by the 2-day hindcast dataset we analysed. We have modified the sentence in question to read: "Although there are no verifying observations for the model timestep data that cover comparable spatial and temporal domains . . . ."

p. 14, L33-34: "Fig. 4a" and "Fig. 4b" should be "Fig. 7a" and "Fig. 7b", respectively.

We agree with the reviewer. We have corrected this error by replacing "Fig. 4a" and "Fig. 4b" with "Fig. 7a" and "Fig. 7b", respectively.

p. 14, Section 4 Discussion: For understanding of properties of cumulus convection schemes, single column models (SCM) have been widely used. Especially, SCM under a radiative-convective equilibrium (RCE) condition is a useful framework for understanding the timestep behaviors. For example, Satoh and Hayashi (1992, J. Atmos. Sci.), Takata and Noda (1997, J. Meteor. Soc. Japan) for SCM in RCE. Please add discussions on the above aspects of using a SCM for understanding of intermittency.

We agree with the reviewer. We have added a short paragraph to the Discussion section of our revised manuscript (second paragraph of Section 4), in which we discuss the potential use of single-column model simulations to investigate the causes of undesirable intermittency in simulated precipitation, including the references that the reviewer pointed out. However, SCM simulations can only address errors in the sub-

gridscale physics, not errors in the coupling between that physics and the resolved dynamics. Hence, we have added the following sentence to the revised manuscript: "Model development efforts to reduce or remove undesirable intermittency may involve single-column model experiments, in which the effects of changes in sub-gridscale physics can be isolated from feedbacks through the resolved dynamics (e.g., Satoh and Hayashi, 1992; Takata and Noda, 1997; Woolnough et al., 2010), although we stress that physics–dynamics coupling may have a substantial effect on the model behaviors and diagnostics presented here."